# A tunable autonomous RNA-fueled micro-engine

Kun Wang [1] ✉, Wenjun Chen[1], Buming Guo [1], Qiuyan Huang[2], Guolong Zhu [3], Heng Ni[1], Lei Zhang[1], Melanie Perez[3], Ruojie Sha [2], Nadrian C. Seeman [2] & Paul M. Chaikin [1] ✉

Autonomous molecular machines capable of converting chemical energy into mechanical motion are foundational components for synthetic nanoscale systems. Inspired by biological motors, we report the construction of a tunable, RNA-fueled DNA origami engine that drives the cyclic movement of a 500 nm-diameter particle at the microscale. The engine operates via sequential RNA–DNA hybridization and enzymatic cleavage by RNase H, enabling reversible switching between folded and unfolded conformations without external intervention. By modulating RNA and enzyme concentrations and controlling temperature, we achieve tunable switching kinetics, with transition periods as short as ~10 s. Kinetic modeling reveals that the folding pathway is governed by both productive RNA binding and the enzymatic clearance of misfolded intermediates, while unfolding is primarily controlled by RNase H activity. Since the RNA fuel binds specifically to the DNA strands, each engine is addressable simply by changing the sequences. This work demonstrates a programmable, self-resetting molecular actuator and offers a blueprint for building more complex nanomechanical systems with forces and energies comparable to molecular motors.

Energy-driving units are fundamental components of complex mechanical systems, as they transform chemical or physical energy into controlled mechanical motion[1–7]. In living organisms, this role is elegantly achieved by various molecular motors powered by adenosine triphosphate (ATP), which drive essential biological processes such as muscle contraction, the movement of cilia and flagella, and cytoskeletal dynamics during cell division and intracellular transport[8–10]. These natural motors operate with remarkable efficiency, directionality, and adaptability, offering a powerful paradigm for synthetic analogs. Inspired by these biological systems, one promising direction in nanotechnology and synthetic biology is the construction of artificial nanoscale engines that can mimic the function of natural molecular machines. Achieving this would not only deepen our understanding of fundamental molecular mechanisms but also pave the way for novel applications in targeted drug delivery[11], biosensing[12], molecular robotics[13], and reconfigurable materials[14,15]. Over the past

two decades, significant progress has been made in building artificial devices—such as pumps[16], walkers[17], rotors[18], and switches[19]—using DNA nanostructures[20], synthetic polymers[21], and proteins[22]. However, most of these systems are either externally actuated or operate in non-autonomous modes.

A key challenge in the development of artificial engines lies in integrating autonomous energy conversion with continuous mechanical output. Compared to their biological counterparts, artificial systems often struggle to generate sufficient force, sustain high-speed operation, or function efficiently at large scales. Thus, the design of robust, self-sustained energy-driving units that can power downstream mechanical functions at the microscale is a nontrivial problem that continues to attract attention. Salaita and co-workers established RNA-fueled DNA rolling motors in which RNase H turnover on RNA-patterned surfaces sustains directed motion[23]. They subsequently adapted this chemistry to DNA-origami–based motors that exhibit

[1]Department of Physics, New York University, New York, NY, USA. [2]Department of Chemistry, New York University, New York, NY, USA. [3]Department of Chemistry, Biochemistry and Physics, Fairleigh Dickinson University, Madison, NJ, USA. ✉e-mail: kunwangneu@gmail.com; chaikin@nyu.edu

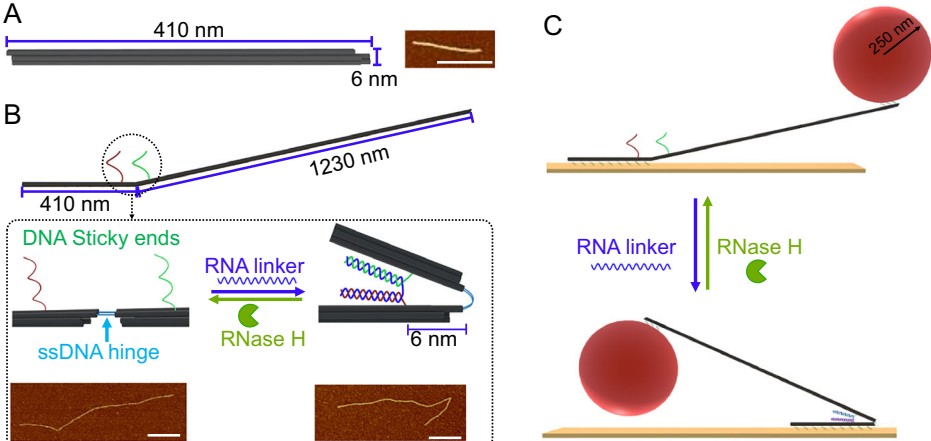

**Fig. 1 | Design and working principle of the RNA-fueled DNA origami engine. A** Schematic of the 6-helix bundle DNA origami (6 HB, ~410 nm) and AFM image of the 6HB (scale bar: 400 nm). **B** Schematic of the engine structure composed of four 6HB DNA origami rods. The first and second rods are connected via two ssDNA hinges that provide mechanical flexibility and elastic restoring force, while the second–third–forth rods are rigidly connected via six dsDNA bridges (Supplementary Fig. 2). The RNA-fueled switching module is located at the junction of the first and second 6HB. Close-up schematic of the hinge region: two sticky ends on the first and second 6HB hybridize with an RNA linker to form a U-shaped structure, maintaining the engine in a folded state. RNase H specifically recognizes and cleaves the DNA/RNA duplex, releasing the constraint and allowing the hinge to open. AFM image of the unfolded and folded conformation of the engine (scale bar: 400 nm). **C** Schematic illustration of the engine construct. A 500nm-diameter particle is tethered to the distal end of the fourth 6HB bundle, allowing optical tracking of switching events. The first 6HB is equipped with 53 single-stranded sticky ends, which can hybridize with a substrate functionalized with BSA–biotin, streptavidin, and biotin–DNA (see Supplementary Figs. 1 and 13 for details). RNA hybridization shifts the engine into the folded state; RNase H cuts the RNA and resets the engine to the unfolded state.

ballistic translocation over micrometer distances at ~nm·s⁻¹ [24]. Collectively, these studies show that coupling hybridization with enzymatic recycling can power mesoscale transport, yet current implementations still depend on prepatterned, track-like surfaces. Here, we introduce a fully automatic DNA engine powered by RNA and RNase H reaction. This design utilizes DNA origami to prepare a lever arm, employing short single-stranded DNA as a hinge spring. We strategically positioned well-designed sticky ends near the hinge vertex to promote efficient RNA linker binding. When the RNA linker binds to the DNA sticky ends, the nanoscale movement is converted into micrometer-scale movement through the lever arm. The engine stores energy in a bendable spring structure by hybridizing DNA with RNA. Then, RNase H cleaves the RNA, releasing the stored energy. With this engine, we are able to autonomously drive the motion of a 0.5 μm-diameter sphere, completing conformational cycles with a period of approximately 10 s. This RNA-fueled engine can serve as a basic unit for driving more complex nanomachines and offers possibilities for constructing autonomous DNA machines. Furthermore, each micro-engine can be individually addressable, only activated by the RNA sequence complementary to the ssDNA on the engine hinge.

## Results

### Design and operation principle of the RNA-fueled DNA origami engine

In this work, we designed an RNA-fueled microscale engine using a six-helix bundle (6HB) DNA origami structure as the fundamental building block (Fig. 1A). Each 6HB rod is approximately 410 nm in length. To construct the engine system, four such 6HB units were assembled into a tetrameric configuration through programmable sticky-end hybridization (Supplementary Figs. 1 and 2). The assembly features distinct mechanical properties across different segments. The first and second 6HB are connected via two short single-stranded DNA (ssDNA) linkers, which serve as flexible hinges, providing the necessary elastic restoring force for mechanical actuation. In contrast, the second, third, and fourth 6HBs are rigidly connected using six double-stranded DNA (dsDNA) bridges to ensure structural stability. The active switching region of the engine is localized at the junction between the first and

second 6HB (Fig. 1B). Two single-stranded DNA sticky ends protrude from the first and second 6HB, which can hybridize with an RNA linker strand of a specific sequence to form a looped "U"-shaped conformation. This conformation maintains the engine in a folded state. Upon hybridization, the RNA linker forms a DNA/RNA duplex, which is recognized and cleaved by the enzyme RNase H. RNase H only degrades the RNA strand when it is hybridized with complementary DNA, and does not act on free RNA strands. Once RNase H cleaves the RNA linker, the mechanical constraint is released. The elastic potential stored in the ssDNA hinge spring then drives the system into an extended, unfolded conformation. The RNA/DNA hybrid duplex can generate hybridization energies of approximately ~8 $k_B$T per base pair [25–27], which is comparable to the free energy released during ATP hydrolysis ( ~ 20 $k_B$T per molecule [28,29]). Moreover, while ATP hydrolysis in protein motors such as myosin yields mechanical forces on the order of ~4pN [30,31], the formation of RNA/DNA duplexes can produce forces up to ~15pN [32–34], indicating that nucleic acid hybridization can serve as a similarly effective energy source for driving mechanical motion. This mechanism enables autonomous, fuel-driven motion: in the presence of both RNA linkers and RNase H, the engine repeatedly undergoes switching cycles, converting chemical energy stored in the RNA fuel into mechanical motion (Fig. 1C). The process is enzyme-gated, mimicking key features of natural molecular motors.

### Validation of the RNA-fueled engine mechanism and real-time observation

In our previous study, we established that mechanical strain—specifically, hinge-induced tension—can modulate the melting temperature of DNA duplexes [35]. To ensure robust operation, we systematically validated several key design parameters of the engine system. First, to ensure RNase H maintains its functional selectivity, under our experimental conditions, we performed PAGE assays (Supplementary Fig. 3), confirming that it selectively cleaves RNA only when hybridized with complementary DNA, consistent with its known biochemical behavior. Second, to ensure that the RNA linker can effectively maintain the engine in the folded state against the restoring force of the ssDNA hinge spring, we screened RNA linkers of various lengths

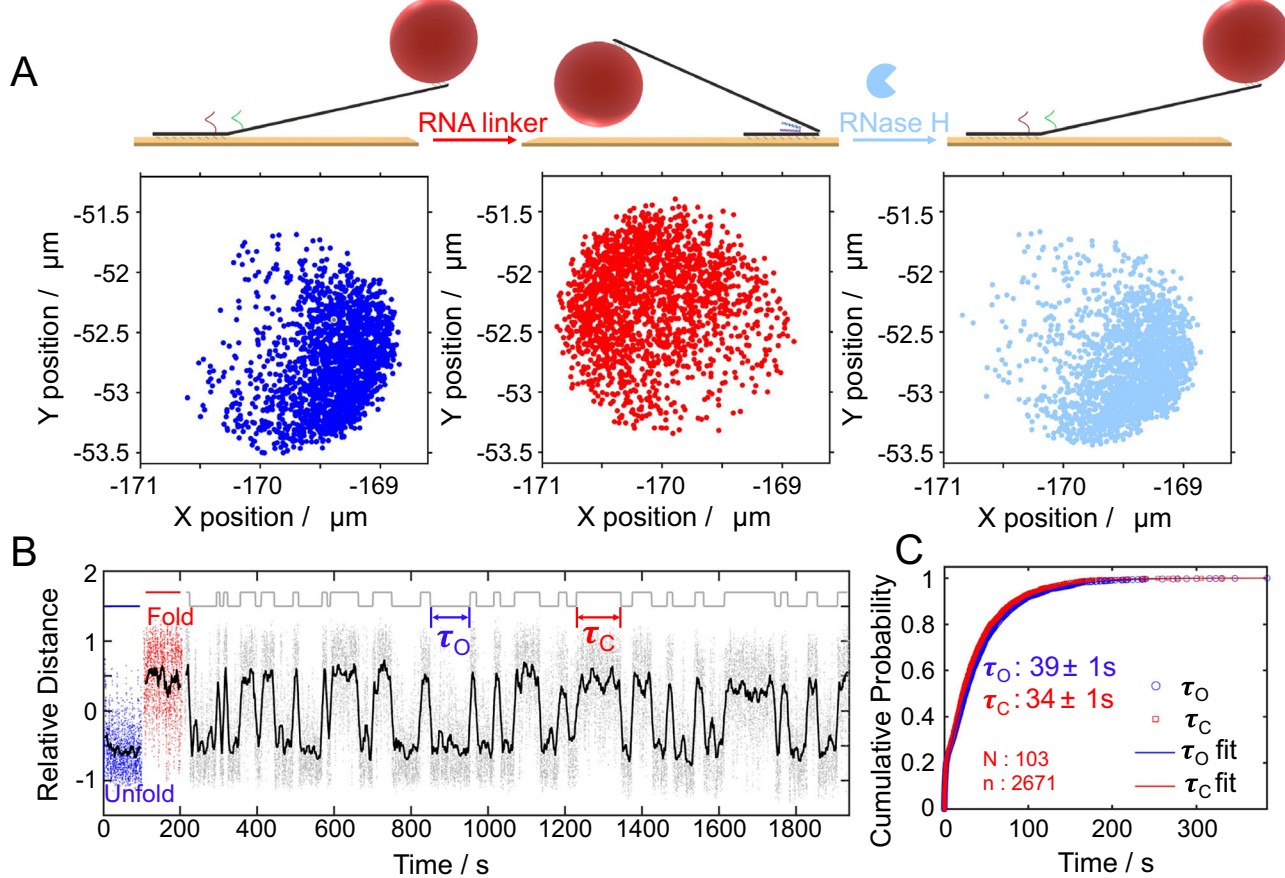

**Fig. 2 | Experimental verification of the engine's switching behavior and dwell time analysis. A** Representative 2D particle tracking data under different conditions: left (blue dots)−engine remains in the unfolded state in the absence of RNA/RNase H; middle (red dots)−engine folds upon addition of RNA linker; right (light blue dots)−engine unfolds again after addition of RNase H, which cleaves the RNA at the DNA/RNA duplex region. **B** Time series of particle relative position (Supplementary Fig. 5), showing autonomous switching behavior in the presence of both RNA (500 nM) and RNase H (80 nM). Raw trajectory (gray dots) and smoothed curve (black line) indicate transitions between folded and unfolded states. The top binary trace represents the idealized conformational state of the engine, assigned as either folded or unfolded based on threshold analysis of the particle's position. Each upward or downward step corresponds to a transition between the two states. Arrows indicate dwell times in folded (closed, $\tau_C$, red) and unfolded (open, $\tau_O$, blue) states. **C** Cumulative distribution of dwell times fitted with a double-exponential model to extract $\tau_O$ and $\tau_C$. (N is the particle number and n is switching event number). The $\tau_O$ and $\tau_C$ are expressed as mean +/− standard error based on 1000 bootstrap resamples. Data were collected at 37 °C with a frame rate of 20 frame per second (FPS). Source data are provided as a Source Data file.

(Supplementary Fig. 4). We found that linkers longer than 32 nucleotides provided sufficient force to overcome the hinge spring's elastic tension and stabilize the folded conformation. A 32-nt RNA linker was selected as the optimal compromise between stability and responsiveness, since longer linkers increased the time required for enzymatic cleavage and system resetting.

To visualize engine operation in real time, we attached a fluorescently labeled polystyrene colloid (500 nm in diameter, FITC- dyed) to the distal end of the moving arm and tracked its two-dimensional position under various conditions using optical microscopy. In the absence of RNA and RNase H, the engine remained in the unfolded conformation due to the hinge spring's force. As shown in Fig. 2A (left panel), the particle diffused within a constrained area, reflecting thermal fluctuations around the unfolded state. Upon addition of the RNA linker, the sticky ends hybridized with the RNA strand, forming a duplex that pulled the engine into the folded conformation. This motion was accompanied by a clear displacement of the particle to the opposite side of the diffusion constrained region (Fig. 2A, middle panel). After removing excess RNA and subsequently introducing RNase H, the particle shifted back to its original location (Fig. 2A, right panel), indicating that RNA cleavage by RNase H successfully released the folding constraint, allowing the spring force to return the engine to its unfolded configuration.

Following stepwise validation of the hybridization, cleavage, and mechanical response processes, we simultaneously introduced RNA and RNase H into the system. This resulted in sustained, fuel-driven operation, wherein the engine continuously cycled between folded and unfolded states, driven by successive rounds of RNA hybridization and enzymatic cleavage (Fig. 2B). To quantitatively analyze the engine dynamics, we converted the 2D particle trajectory into a one-dimensional relative distance trace. In the presence of both RNA linker and RNase H, the engine underwent continuous autonomous switching between folded and unfolded states. The two states show partial overlap due to the flexible DNA arms, but running average smoothing reveals clear transitions. From the smoothed trajectory, we extracted the dwell times of each conformational state (Supplementary Fig. 5). Supplementary Movie 1−3 further show that the particle mainly diffuses around two separate position ranges corresponding to the unfolded and folded configurations; with RNA and RNase H present, it autonomously switches between these two ranges. Although the absolute displacement looks small in the raw videos, aligning the tracks and converting to a relative-distance coordinate makes the state transitions and trajectory clear. As shown by the log-survival comparisons and information-criterion scores, a double -exponential survival provides a markedly better description than a single exponential (Supplementary Fig. 6 and Supplementary Text 1). We therefore used

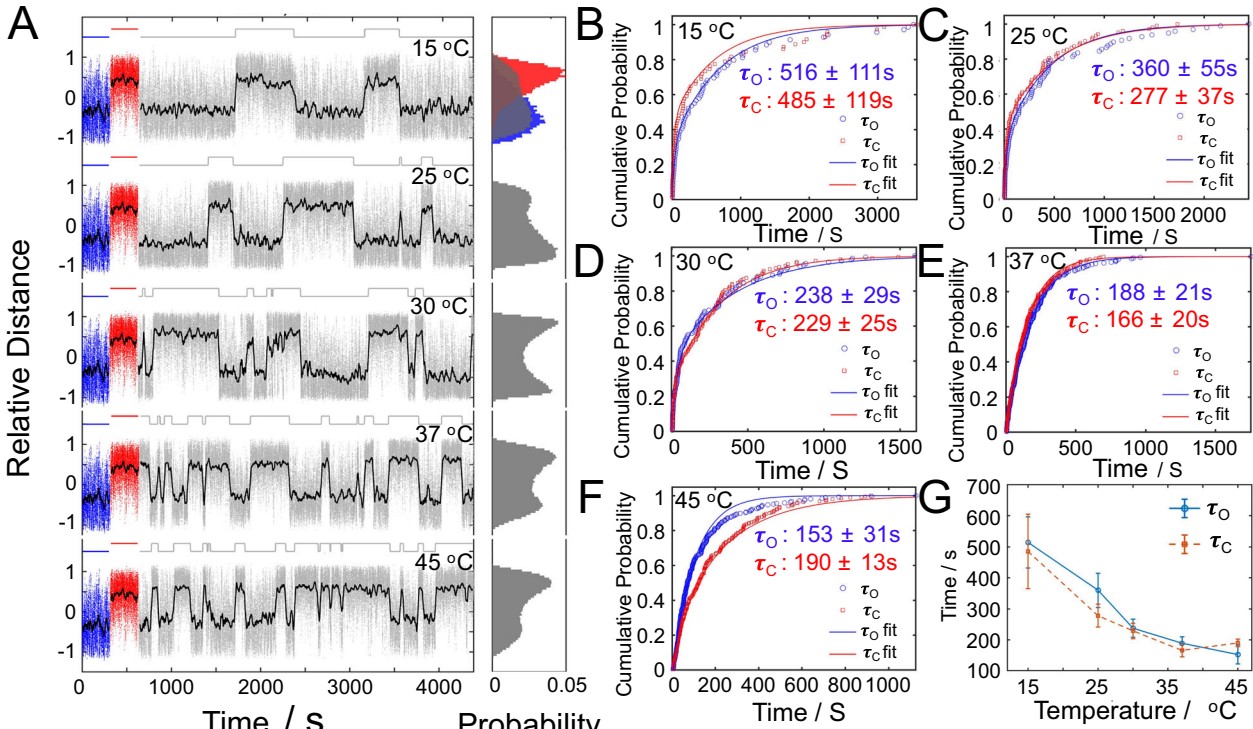

**Fig. 3 | Temperature dependence of the engine's switching dynamics.**
**A** Representative trajectories of the RNA-powered engine recorded at various temperatures (15–45 °C), illustrating transitions between the folded and unfolded states (RNA: 5 nM, RNase H: 11 nM). Black lines represent trajectories derived from a 20-s running average of the bead's position. The leftmost colored segments denote the control states: unfolded (blue) for the engine alone and folded (red) for the engine with RNA linker only. Histograms on the right display the corresponding relative position distributions for each trajectory. **B**–**F** Cumulative distribution functions of the unfolded dwell times ($\tau_O$, blue) and folded dwell times ($\tau_C$, red) at different temperatures. Solid curves represent double-exponential fits used to extract characteristic time constants. N is the particle number and n is switching event number. **G** Extracted $\tau_O$ and $\tau_C$ values as a function of temperature. Both time constants decrease with increasing temperature, indicating that hybridization and enzymatic cleavage rates are thermally accelerated. The $\tau_O$ and $\tau_C$ are expressed as mean +/− standard error based on 1000 bootstrap resamples. Data were collected with a frame rate of 20 FPS. Source data are provided as a Source Data file.

the double-exponential model throughout. A statistical analysis of over 1000 transition events yielded the mean dwell times for the unfolded ($\tau_O = 39 \pm 1$ s) and folded ($\tau_C = 34 \pm 1$ s) states (Fig. 2C), obtained by double-exponential fitting of the cumulative distribution of state lifetimes. These results confirm the autonomous, enzyme-mediated switching behavior of the engine and provide kinetic insight into its operation. As controls, in the absence of both RNA and RNase H the device remained in the unfolded state throughout entire recordings (≥ 30 min) and switched to the folded plateau only upon subsequent RNA addition (Supplementary Fig. 7A); furthermore, a scrambled RNA linker did not induce folding (Supplementary Fig. 7B). The scrambled linker preserves the length but was sequence-shuffled to eliminate complementarity to both sticky ssDNA ends (Supplementary Table 1). Together, these results support sequence-specific actuation and indicate that each micro-engine is individually addressable, activating only in response to the RNA sequence complementary to the DNA sticky ends.

### Temperature-dependent kinetics of the RNA-fueled engine
To investigate how temperature affects the performance of the RNA-fueled DNA engine, we monitored trajectories across a range of temperatures (15–45 °C). As shown in Fig. 3A, the engine exhibited stochastic transitions between the folded and unfolded states at all tested temperatures, with transition frequencies increasing markedly at higher temperatures. Although the data exhibit considerable noise, the relative position distributions still display two well-separated states, demonstrating that the engine undergoes autonomous two-state switching driven by RNA hybridization and enzymatic cleavage. We analyzed the mean dwell times in the unfolded ($\tau_O$) and folded ($\tau_C$)

states across multiple trajectories. Cumulative distribution functions (Fig. 3B–F) were fitted using a double-exponential model, yielding characteristic time constants for each state. Notably, both $\tau_o$ and $\tau_C$ decreased significantly with increasing temperature, as shown in Fig. 3G. This trend suggests that both the RNA hybridization process, which drives the folding transition, and the RNase H-mediated cleavage, which facilitates unfolding, are accelerated at elevated temperatures. Notably, $\tau_C$ at 45 °C was slightly bigger than at 37 °C, consistent with the known optimal activity of RNase H near physiological temperature; enzymatic cleavage becomes less efficient at higher temperatures due to potential thermal denaturation or decreased enzyme stability[36,37]. These results confirm that the engine's switching is thermally tunable, with enzymatic and hybridization steps contributing temperature dependencies.

### Effect of RNA and enzyme concentration on engine switching kinetics
To further investigate the determinants of the engine's switching behavior, we investigated how varying the RNA linker concentration influences the dwell times of the two conformational states. By definition, $\tau_C$ corresponds to the time the engine remains in the folded state, which should primarily depend on the RNA cleavage rate catalyzed by RNase H. In contrast, $\tau_O$ reflects the time spent in the unfolded state, which should be dictated by the hybridization rate of the RNA linker to the DNA sticky ends. To validate this mechanistic distinction, we conducted two sets of experiments. In the first, we fixed the RNase H concentration and systematically varied the RNA linker concentration (Fig. 4A–E). Under these conditions, $\tau_C$ remained relatively stable, confirmed that cleavage kinetics are governed by enzyme

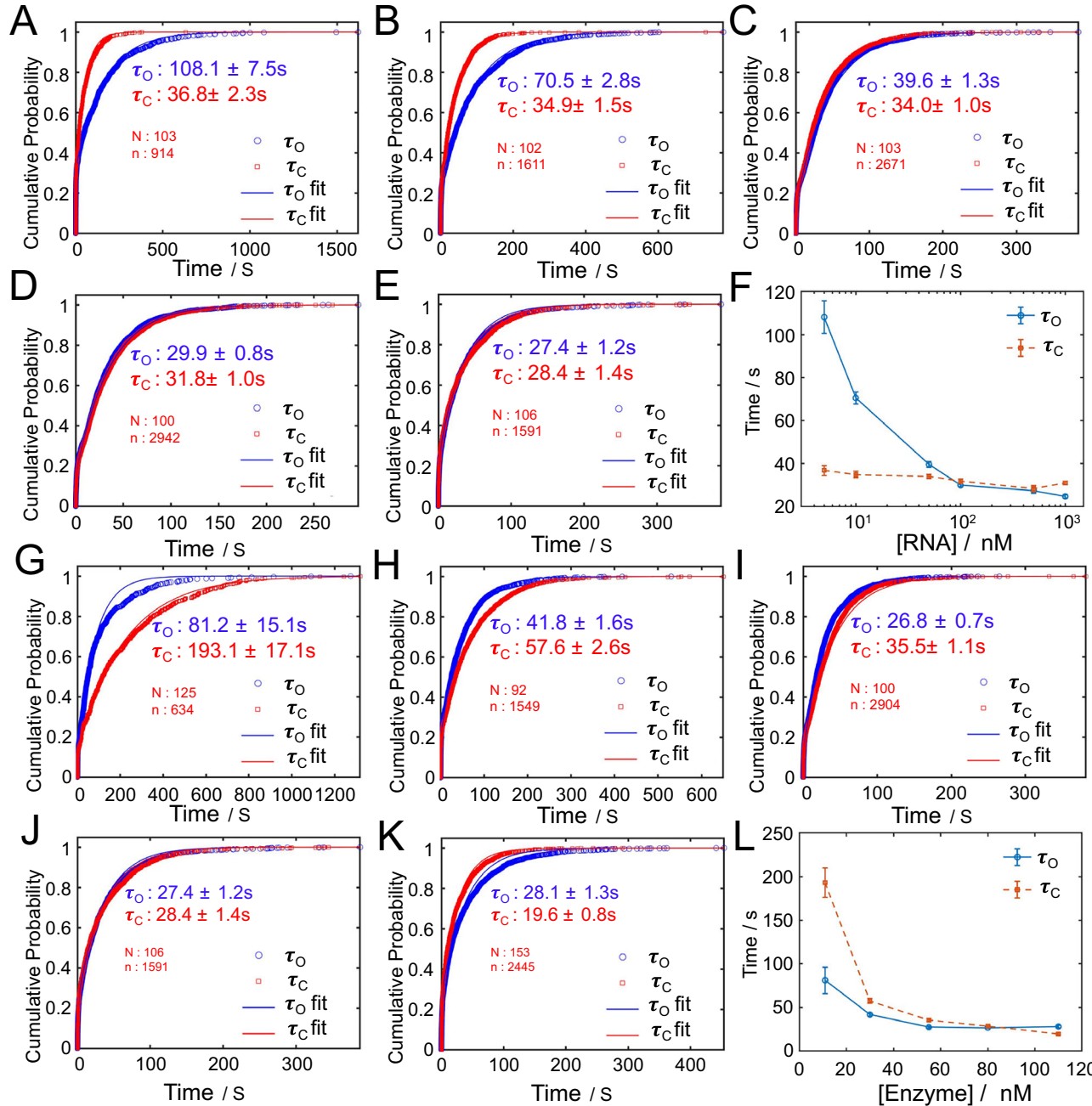

**Fig. 4 | Modulation of dwell times by varying RNA and RNase H levels.**
**A–E** Cumulative probability plots of dwell times under varying RNA concentrations (5, 10, 50,100 and 500 nM) with RNase H concentration held constant (80 nM). Blue: $\tau_O$ (time remains at unfold state); Red: $\tau_C$ (time remains at fold state). Each dataset was fitted using a double-exponential decay model. **F** Plot of $\tau_O$ and $\tau_C$ vs. RNA concentration (log scale). **G–K** Cumulative probability plots of dwell times under varying RNase H concentrations (11, 30, 55, 80 and 110 nM) with RNA concentration held constant (500 nM). Blue: $\tau_O$; Red: $\tau_C$. Each dataset was fitted using a double-exponential fit. **L** Plot of $\tau_O$ and $\tau_C$ vs. enzyme concentration. The $\tau_O$ and $\tau_C$ are expressed as mean +/− standard error based on 1000 bootstrap resamples. Source data are provided as a Source Data file.

concentration. Meanwhile, $\tau_O$ decreased as RNA concentration increased, confirming that RNA hybridization is the dominant rate-limiting step for folding in this context (Fig. 4F).

Then, we varied the enzyme concentration while keeping the RNA linker concentration constant. As expected, increasing RNase H concentration significantly reduced $\tau_C$, consistent with accelerated RNA cleavage (Fig. 4G–K). Interestingly, $\tau_O$ also showed a notable decrease, suggesting that RNase H may also influence the folding process−potentially by binding transiently to the RNA linker or assisting in stabilizing the DNA/RNA duplex during hybridization (Fig. 4L). Together, these results suggest that while $\tau_C$ is governed primarily by

enzymatic activity, $\tau_O$ is influenced by both RNA availability and potentially enzyme-mediated facilitation of hybridization.

To understand the $\tau_O$ (unfolded dwell time) and the $\tau_C$ (folded dwell time) dependance on RNA concentration and enzyme concentration, we developed a kinetic model that captures the key molecular steps involved. For the unfolding process, the kinetics are primarily governed by enzymatic cleavage (Supplementary Text 2 and Supplementary Fig. 8). After extracting the kinetic constants from our data, we compared the fitted parameters for the unfolding process with values reported in the literature[38,39]. The estimated rates for dissociation ($k_d$) are on the same order of magnitude as those previously

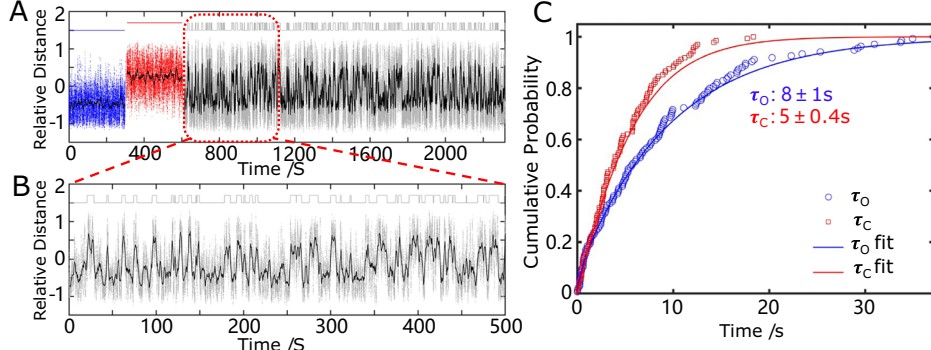

**Fig. 5 | Real-time monitoring of the engine under high-concentration conditions. A** Time series of the particle's relative position. Blue dots represent the control unfolded state, and red dots indicate the control folded state. The gray trace shows the raw trajectory of an engine undergoing stochastic folding and unfolding in the presence of RNA (10 μM) and RNase H (550 nM), with the smoothed black curve highlighting transitions between states (2 s running average). The binary trace on top denotes the idealized conformational states assigned as folded or unfolded based on position thresholding. **B** Zoomed-in view of the first 500 s of the raw trajectory shown in **A**. **C** Cumulative probability distributions of dwell times in the unfolded ($\tau_O$) and folded ($\tau_C$) states, fitted with a double-exponential model to extract characteristic lifetimes. Data were collected at 37 °C with a frame rate of 200 FPS. Source data are provided as a Source Data file.

reported for RNase H in similar systems. However, the extracted turnover rate ($k_{cat}$) is substantially higher than typical literature values, which generally report cleavage rates around -1 s$^{-1}$ (Supplementary Fig. 9). This discrepancy likely arises from a key mechanistic difference: in our system, partial RNA cleavage is sufficient to trigger engine activation, whereas reported $k_{cat}$ values usually reflect the rate of complete RNA digestion. This interpretation is also supported by native gel electrophoresis results (Supplementary Fig. 4). Additionally, the observed enzyme association rate constant ($k_a$) is lower than previously reported values, likely due to structural constraints in our system. In the folded state, the RNA/DNA hybrid is partially buried within the folded engine, which may hinder enzyme accessibility. This spatial hindrance slows down enzyme binding, resulting in a reduced apparent $k_a$.

For the folding process, the engine begins in an unfolded state and requires sequential binding of RNA linkers to initiate productive folding (Supplementary text 3 and Supplementary Fig. 10). However, if two RNA strands bind simultaneously, the system can become trapped in a misfolded intermediate that cannot proceed to folding without intervention. In such cases, RNase H is required to cleave the bound RNA and reset the system to the unfolded state. Our kinetic model captures both productive and unproductive folding pathways, incorporating RNA binding, enzymatic interactions, and the conformational transitions that ultimately lead to folding. The model predicts the dwell time in the unfolded state prior to successful folding (Supplementary Fig. 11). This framework explains why the unfolded dwell time ($\tau_O$) is sensitive to both RNA and enzyme concentrations: RNA concentration controls the rate of hybridization, while enzyme concentration determines the efficiency of clearing misfolded intermediates. High RNA concentrations accelerate the initiation of folding, whereas sufficient enzyme activity ensures rapid removal of incorrect configurations, thereby enabling repeated folding attempts.

As shown in Fig. 2A, the moving arm of our system is not sufficiently rigid to produce two well-separated conformational states. This structural flexibility leads to significant overlap in positional signals and elevated noise levels, making state assignment challenging. To mitigate this, we applied a running average to smooth the trajectory for downstream analysis. However, this approach introduces an inherent limitation: short-lived transition events are often attenuated or entirely missed, preventing accurate detection of rapid switching dynamics. Despite this, under conditions of high RNA and enzyme concentrations and with the aid of a high-speed camera, we were still able to observe cyclic transitions with cycle times about 10 s (Fig. 5).

The model recapitulates the experimental CDFs for both $\tau_O$ and $\tau_C$ under high concentrations of RNA and RNase H (Supplementary Fig. 12), supporting its applicability in capturing the system's key kinetic features. The analysis reveals that $\tau_C$ can be effectively reduced by increasing enzyme concentration alone. In contrast, modulation of $\tau_O$ requires simultaneous adjustment of both RNA and enzyme levels, reflecting its dependence on both hybridization and cleavage steps. As a possible improvement, label-free readouts (RICM or dark-field) are compatible with our assay and may enhance localization precision and state separability with only optical changes, and we therefore note them as promising options for future iterations[40,41].

## Discussion

We have developed a tunable, autonomous RNA-fueled engine capable of driving the cyclic movement of a 500 nm-diameter particle over microscale distances. The engine is powered by two key biochemical processes: RNA–DNA hybridization, which drives conformational folding, and RNA cleavage by RNase H, which resets the system. Through the well design of a ssDNA hinge spring and the control of biochemical inputs, we demonstrate that the engine can operate continuously and switch between folded and unfolded states without external intervention.

The engine's dynamic behavior is highly tunable. By adjusting the concentrations of RNA and enzyme, as well as the operating temperature, we can modulate the switching kinetics over a wide range. With high RNA and RNase H concentrations and the use of high-speed imaging—we observed robust cyclic transitions with periods as short as -10 seconds. Mechanistic modeling and kinetic analysis reveal that the folding process is regulated by a balance between productive RNA binding and the clearance of misfolded intermediates, while the unfolding process is primarily governed by enzymatic cleavage. The system's responsiveness to environmental cues and its ability to reset misfolded states via enzymatic activity highlight its potential as a self-regulating, reusable nanomechanical actuator. Moreover, each engine has its own unique code, the sequence of the ssDNA's at the hinge vertex. If each ssDNA pair is the same, adding the complementary RNA strand activates every engine in the system. If the system has two types of machines, each with a different ssDNA sequence then each can be activated individually by adding its own complementary RNA. This suggests more complex arrangements where transcription can produce the complementary RNA for each type of engine. The transcription can be controlled by a promotor that activates transcription in response to a certain stimulus. Thus, it will be possible to construct systems with many sensors and responses.

Overall, our work establishes a platform for RNA-fueled engine and provides design principles for building more complex, chemically powered nanomachines capable of performing autonomous, directional, and cyclic mechanical tasks. Looking forward, this system offers a foundation for constructing next-generation nanoscale devices such as molecular transporters, logic-controlled actuators, or synthetic molecular assemblers. Moreover, by integrating additional regulatory modules or coupling with sensing elements, such engines may find applications in biosensing, smart drug delivery, or adaptive biomaterials that respond dynamically to their biochemical environment.

## Methods

### Preparation of the DNA origami

DNA origami structures were designed in caDNAno[42] (file provided in Supplementary Data 1). Full sequences are detailed in the Supplementary Information. The DNA origami preparation began with DNA origami buffer (1× TAE with 10.5 mM $Mg^{2+}$). The assembly solution consisted of 200 μL total volume, containing M13mp18 scaffold DNA at a concentration of 10 nM, and each staple strand at 100 nM. The annealing process was carried out in a thermal incubator following a multi-step temperature program: the mixture was first incubated at 70 °C for 30 min, followed by a slow cooling ramp from 70 °C to 28 °C at a rate of 10 °C per hour. The sample was then held at 28 °C for 5 h to promote complete folding, and subsequently cooled to 4 °C at a rate of 60 °C per hour. Finally, the sample was held at 4°C for storage until further use. Purification of the folded DNA origami structures was performed using a 100 kDa Amicon centrifugal filter. First, 400 μL of the origami solution in buffer was added to the filter unit and centrifuged at 2200 × g for 12 min. This was followed by three sequential washes: in each step, 400 μL of DNA origami buffer was added to the filter and centrifuged under the same conditions. After the final wash, the filter unit was inverted into a clean tube and centrifuged at 2400 × g for 3 min to recover the purified DNA origami.

### Preparation of DNA origami-colloids

For the sequences and structural details of the six-helix bundle (6HB) origami, refer to Supplementary Tables 2–4 and Supplementary data 2–5, Supplementary Figs. 1 and 2. As shown in Supplementary Fig. 1, four 6HB structures (6HB-1, 6HB-2, 6HB-3, and 6HB-4) were mixed at equimolar concentrations (5:5:5:5 nM) to assemble a tetramer structure. This mixture was subjected to standard DNA origami annealing conditions as described previously[35]. The successful formation of the 6HB-tetramer structure was confirmed by atomic force microscopy (AFM), as shown in Fig. 1B.

To prepare the 6HB-tetramer–Colloid complex, polystyrene beads were first functionalized with DNA brushes (designated as DBCON-20T-Colloid; see Supplementary Table 4 and Supplementary Fig. 1), followed by assembly with the 6HB-tetramer, as illustrated in Supplementary Fig. 1. The polystyrene colloidal beads (500 nm diameter, 10% w/v) were modified via azide–alkyne cycloaddition (click chemistry) using the following procedure: 30 μL of beads were mixed with 70 μL of 1 M tri-block azide-functionalized polymer (polystyrene (3.8 kDa)−PEO (6.5 kDa)−azide), 180 μL of water, 120 μL of tetrahydrofuran (THF), and 3 μL of FITC/toluene solution. The mixture was shaken at 1000 rpm for 5 h. The beads were then washed four times with water and concentrated to 260 μL, yielding polymer-brush-modified colloids.

To functionalize the particles with DNA, 140 μL of the polymer-brush-colloid suspension was mixed with 80 μL of 100 μM DBCON-modified DNA (DBCON-20T-Colloid in Supplementary Table 4), 560 μL of PBS, 40 μL of 1% (w/w) Pluronic F127, and 20 μL of water. This mixture was shaken at 1000 rpm for 48 h. The resulting particles were washed four times with water and concentrated to 320 μL to obtain the final DNA-coated colloids.

Reference particles for tracking (immobilized on the substrate to eliminate drift during microscopy; DBCON-Ref-particle in Supplementary Table 4) were modified using the same DNA functionalization procedure as described for DBCON-20T-Colloid.

To assemble the DNA-origami–Colloid complex, the DNA-coated colloids were incubated with DNA origami in an assembly buffer composed of 1× PBS (155 mM Na+), 10.5 mM MgCl₂, and 0.1% (w/w) Pluronic F127. The components were mixed and gently rotated at room temperature for several hours.

### Preparation of sample cell

The sample cell was assembled in three steps (Supplementary Fig. 13A). First, a glass coverslip (18 × 18 mm, ~0.13 mm thick) and a glass slide were thoroughly cleaned with ethanol and deionized water. Second, a strip of double-sided adhesive tape (~0.1 mm thick) was applied to the glass slide to define the chamber. Finally, the cleaned coverslip was gently adhered to the tape, forming a sealed sample cell with dimensions of approximately 10 × 18 mm and a total chamber volume of about 15 μL.

To ensure specific surface attachment of the RNA-fueled engine, the inner surface of the sample chamber was functionalized with complementary biotin-DNA through a stepwise assembly process (Supplementary Fig. 13B). First, 20 μL of biotin-labeled bovine serum albumin (2 mg/mL) was added and incubated for 15 min to passivate the surface and introduce biotin groups. The chamber was then washed three times with 60 μL of assembly buffer. Next, 30 μL of streptavidin solution (2 mg/mL) was added and incubated for 15 minutes to form biotin–streptavidin linkages, followed by seven washes with 60 μL of assembly buffer. This was followed by the addition of 30 μL of biotin-modified single-stranded DNA (10 μM), which was incubated for another 15 min to allow hybridization with the streptavidin layer, and again washed seven times.

Subsequently, 20 μL of the DNA-origami–Colloid complex was introduced and incubated for 30 min to allow specific hybridization with the surface-anchored DNA. The chamber was washed three times with assembly buffer to remove unbound structures. Next, 20 μL of tracking reference particles was added and incubated for 5 min, followed by three additional washes to remove excess particles. Then, thin inlet and outlet tubes were inserted at both ends of the chamber to allow fluid exchange. Finally, the chamber was sealed using UV-curable adhesive to ensure sample stability during imaging.

### Data recording and particle tracking

The setup is schematized in Supplementary Fig. 13. Experiments were performed on a Nikon Eclipse Ti inverted fluorescence microscope equipped with a Nikon Plan Apo Lambda 100× oil-immersion objective. Excitation was supplied by a SPECTRA Light Engine (Lumencor) with a Chroma 49002 (ET-EGFP/FITC/Cy2) filter set. Time-lapse fluorescence movies of the RNA-fueled engine were acquired on an Andor Zyla sCMOS camera under Nikon NIS-Elements control at 37 °C (except for Fig. 3). Raw movies were preprocessed in ImageJ with a linear contrast adjustment to enable robust localization. Particle trajectories were then extracted and analyzed in Python (v3.13) using Trackpy (v0.6.1) together with NumPy and SciPy. For each dataset, we first recorded a movie without RNA and without RNase H to define the unfolded state. We then introduced RNA, waited 10 min, and recorded a second movie to define the folded state. The sample was rinsed with buffer. We subsequently added RNA and RNase H at the indicated concentrations and recorded the experimental movies used for analysis. As shown in Supplementary Fig. 5. We determined the maximum-likelihood position of the unfolded state 'u' from the control without RNA linker and the maximum-likelihood position of the folded state 'f' from the control with RNA linker. At time '$t$', we computed the distances $\rho_u(t)$ and $\rho_f(t)$ from the colloid to the u and f positions and the distance $\rho_{fu}$

between f and u. The state is quantified by $(\rho_f(t)^2 - \rho_u(t)^2)/(\rho_{fu}^2)$. Values below zero indicate the unfolded state, and values above zero indicate the folded state.

## Polyacrylamide gel and agarose gel electrophoresis

To confirm RNase H cleavage, 10% native polyacrylamide gel electrophoresis (PAGE) was performed. A GeneRuler Ultra Low Range DNA Ladder was used as the molecular marker. Electrophoresis was carried out at 10 V/cm at room temperature, and the gel was subsequently stained with ethidium bromide (EB) for visualization (Supplementary Fig. 3).

For structural analysis of DNA origami samples, 0.8% agarose gels were prepared in 1× TAE/Mg$^{2+}$ and cast in standard trays. Samples (DNA origami: RNA linker = 1:10, molar) were incubated for 10 min at room temperature prior to loading. Five microliters of each sample were mixed with 1 μL of a non-denaturing loading dye (50% glycerol with trace amounts of bromophenol blue and xylene cyanol FF in 1× TAE/Mg$^{2+}$). Electrophoresis was performed at 4–5 V cm$^{-1}$ for 3 h at 48 °C. Gels were stained with ethidium bromide (EB), rinsed with deionized water, and imaged under UV transillumination (Supplementary Fig. 4).

## Reporting summary

Further information on research design is available in the Nature Portfolio Reporting Summary linked to this article.

## Data availability

The data supporting the findings of the study are available in the article and its Supplementary Information. All data are available from the corresponding author upon request. Source data are provided with this paper. Source data is available for Figs. 2B–C, 3B–G, 4 and 5C in the associated source data file. Uncropped original gel images for Supplementary Figs. 3 and 4 are also included. Source data are provided with this paper.

## Code availability

The code used for particle tracking is based on the publicly available Trackpy package (https://github.com/soft-matter/trackpy). The custom Python scripts developed for kinetic analysis, including double-exponential weighted fitting and model simulations, are available in a public GitHub repository (https://github.com/kw2556nyu/A_Tunable_Autonomous_RNA_Fueled_Micro_Engine_Analysis_Code.git) and archived with the permanent identifier https://doi.org/10.5281/zenodo.18173464. The analysis relies on standard open-source libraries (NumPy, SciPy, Matplotlib) as specified in the repository documentation.

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

## Acknowledgements

This research received primary support from the U.S. Department of Energy under Award DE-SC0007991 (P.M.C., N.S., R.S., K.W.), which covered project conception, planning, and data analysis. The experimental implementations were supported by DOE Award DE-SC0020971 (K.W.). Computational modeling efforts were funded by the National Science Foundation through the NSF-BSF Organization Far From Equilibrium Program (GRANT NO 2414721) and the ISS: GOALI initiative (NSF Grant No. 11832291) (B.G., W.C.). Early-stage design efforts were supported by the Center for Bio-Inspired Energy Sciences (CBES), an Energy Frontier Research Center sponsored by the U.S. DOE Office of Science, Basic Energy Sciences (Award DE-SC0000989) (G.Z.). Additional funding was provided by the Office of Naval Research (ONR Grant N000141912596) and NSF CCF-2106790 (R.S., N.C.S.), and P.M.C. acknowledges support from the Simons Foundation (Award No. 7200138).

## Author contributions

Project conceptualization was led by P.M.C., N.C.S., R.S., and K.W. Sample fabrication and preparation were performed by K.W. Experimental procedures and data analysis were conducted by K.W., W.C., G.Z., Q.H., B.G., P.M.C., and N.C.S.; Overall project oversight and coordination were carried out by P.M.C. and N.C.S. The initial manuscript draft was prepared by P.M.C. and K.W. The revision experiments were conducted by G.Z., M.P., and L.Z.

## Competing interests

The authors declare no competing interests.
