## [Transparent Peer Review file · Nature Communications]

A Tunable Autonomous RNA-Fueled Micro- Engine

Corresponding Author: Dr Kun Wang

Version 0:

Reviewer comments:

Reviewer #1

(Remarks to the Author)

General remarks:

This article by Wang & Chaikin et al reports the design of a DNA origami hinge machine that switches between folded and unfolded conformations upon RNA hybridization and enzymatic cleavage by RNase H respectively, driving micrometer-scale movements of a 500 nm-diameter particle. The hinge machine is comprised of four 6-helix bundle (6HB) units assembled into two arms: a short, immobilized arm and a longer, semi-rigid moving arm. The two arms are connected through a flexible hinge spring composed of two single-stranded DNA domains, and both present sticky ssDNA protrusions close to the hinge. Introduction of an RNA linker that bridges the sticky DNA ends induces hinge closing, whereas RNase H-mediated hydrolysis of the bound RNA releases the stored elastic energy, restoring the origami structure to its unfolded conformation.

In their previous work, Wang and Chaikin et al demonstrated the ability to control the folding and unfolding transitions of an analogous DNA origami construct using a microheater to change the local temperature (Science Advances, 2025). The main difference in the molecular hinge design was that the sticky DNA protrusions on opposing arms could directly hybridize. At lower temperatures, hybridization stabilized the folded state, while higher temperatures induced duplex melting, favoring the open conformation. Each transition was precisely controlled by external input as applying current through the heater enabled rapid equilibration to the programmed temperature within milliseconds.

The new proposed work is an interesting implementation of the RNase H-powered consumption of RNA fuel molecules for driving autonomous motion and manipulation of microscale or nanoscale objects. Unlike the previously reported heat-activated hinge machine, this RNA-fueled machine is fully autonomous. This property also sets this work apart from the class of photoactivated DNA switches, which lack both processivity and autonomy (Willner et al, Angew. Chem. Int. Ed., 2017; Kuzyk et al, Nature Communications, 2016). While the design of a lever arm using DNA origami technology is not itself novel, this work presents some clear scientific merit.

Furthermore, the PAGE assay included to determine the specific activity of RNase H on the selected RNA sequence is valuable as the enzyme is known to exhibit sequence preferences as well as non-specific cleavage with certain sequences (Kielpinski et al, Nucleic Acids Research, 2017, Vol.45, No.22). Hence, validating that RNase H enzyme selectively cleaves RNA substrate in DNA:RNA hybrids employed in the system, while leaving ssDNA and ssRNA intact, was an important and well-justified step. The mechanistic insights and kinetic analysis are also thorough and add value to the paper.

That said, I have a few concerns that require the active consideration of the authors. My comments are listed below:

Major concerns:

1. The authors claim that the hinge machine is an RNA-fueled "motor". However, the DNA machine described in this work behaves more like a molecular switch that transitions between two states, folded and unfolded, rather than a processive motor that continuously converts chemical energy into net mechanical work. While the attached particle should move across micrometer distances, we only observe the endpoints, reinforcing the interpretation of a binary switch.

Based on the extent of motility and type of mechanical movement, the hinge machine reported in this study is more akin to a nano-engine, similar to the one developed by Centola & Famulok et al (Nature Nanotech., 2024) that generates a rhythmic pulsating motion powered by T7 RNA polymerase. The distinction between DNA switches and DNA motors is a critical one, and these terms should not be used interchangeably.

If a switching mechanism, which on its own cannot generate net mechanical work, is incorporated into a motor system that produces net mechanical work over a cycle in the form of translocation along a linear track or across a 2D surface, it would better qualify as a motor. An example is the bar-hinge motor reported by Small et al (New Journal of Physics, 2019). However, the RNase H-activated hinge machine described in this study lacks this property.

2. Building upon my previous comment, I strongly suggest including fluorescence microscopy snapshots of single particles

over time. Real-time fluorescence tracking data revealing particles that dwell in intermediate positions could provide evidence for motor-like behavior. For example, the electric field-driven DNA rotary motor reported by Kopperger & Simmel et al (Science, 2018) was shown at different positions along the circular trajectory in snapshots taken using fluorescence microscopy.

3. A search of the literature quickly reveals that there is important literature on “RNA-Fueled Micro-Motors” that were not cited or mentioned in this report. This needs to be cited: 1) Yehl et al. 2016 Nature Nanotechnology (this paper shows the first RNA fueled micromotor). 2) Bazrafshan et al. 2020 Angewandte Chemie (this paper shows the first RNA fueled origami motor).

4. In Fig. 2A, how do the authors ensure a preferential orientation of hinge machines in their open conformation? The particle distribution appears skewed, with a higher density in the bottom-right region of the field of view. How is the orientation of the origami arms controlled during surface immobilization? One would expect the origami structures to deposit randomly, both in position and orientation, resulting in a more homogeneous distribution across the surface.

5. FITC (Fluorescein isothiocyanate) is used as fluorescent label on the polystyrene colloidal particles for tracking purposes. FITC is widely used for applications that do not require high sensitivity and/or long exposure because it is less bright than other green-fluorescing dyes and it is prone to rapid photobleaching. Photobleaching occurs when the fluorescent molecule is irreversibly degraded by exposure to high-intensity light and loses its ability to fluoresce. There are several alternative dyes with significant greater brightness and photostability, such as Alexa fluor 488. In this study, the experimental design does not impose restrictions related to orthogonal fluorophore requirements, so alternative dyes that are excited or emit at different wavelengths (e.g., ATTO 565, ATTO 647N) could be readily employed.

Do the authors employ any software, such as ImageJ Fiji, to correct for the diminishing signal caused by photobleaching? It is somewhat surprising that the FITC signal remains sufficiently bright given the extended imaging duration (over 30 minutes) and the high acquisition frequency reported in this study.

Given the size of the colloidal particle (500 nm diameter), label-free approaches like reflection interference contrast microscopy (RICM) and dark-field scattering microscopy should be effective for single-particle tracking. RICM was used to track 50-nm gold nanoparticle motors (Bazrafshan et al, ACS Nano, 2021), and dark-field scattering microscopy can track 100-nm gold nanoparticles with 1-ms time resolution and 1-nm localization precision (Harashima et al, Nature Communications, 2025).

6. Although the authors provide convincing data showing that switching kinetics can be tuned by changing key parameters (temperature, RNA concentration or enzyme concentration), the underlying switching behavior may merely reflect the confined diffusion of the attached particle. As a control, time series of particle relative position of individual origami-particle complexes in the absence of both RNA and RNase H should be included. Such data would strengthen the mechanistic interpretation by demonstrating that the hinge remains in the open conformation for the full duration of the timelapse experiment without open/close shifts. It is also possible that the release of elastic energy following the first RNase H cleavage event might be sufficient to generate an oscillatory motion of the lever arm with the colloidal particle.

7. A control experiment using a scrambled RNA linker that fails to bridge the sticky ssDNA ends should be included. Since no strategy to visualize RNA hydrolysis is being used, this control would help confirm that hinge closing arises specifically from RNA hybridization rather than from non-specific interactions.

Minor issues:

1. The DNA origami is most likely designed using caDNAno, as also suggested by the cropped images in Fig. S11.

However, neither the main article nor the Supporting Information cite this widely used, open-source software package (Douglas & Shih et al., Rapid prototyping of 3D DNA-origami shapes with caDNAno, Nucleic Acids Research, 2009).

2. Supplementary figures are not cited in a consecutive order in the main text. For example, S11 is referenced right after S1 (page 2, line 24), and S2 does not appear before S3. I suggest rearranging the SI figures according to the first time they are mentioned in the main text.

3. Suggest adding the fragment sizes in bp for the bands in the DNA ladder in Fig. S3.

4. The screening of RNA linkers of various length via non-denaturing agarose gel electrophoresis in Figure S4 indicates that all RNA linkers from 28 nt to 58 nt produce dual populations of folded and unfolded 6HB dimers, as evidenced by two distinct bands under each condition. This suggests that some 6HB dimers fail to fold into the closed conformation in the presence of the RNA linker. What is the ratio of origami to RNA? How long are they incubated together before gel analysis?

5. Could shorter RNA linkers with higher GC content lead to faster binding/folding rate while maintaining duplex stability?

6. The word “colloid” (not “collid”) is spelled incorrectly in Fig. S1.

7. Double commas on page 3, line 5.

8. There is a typo on page 3, line 15. It should be “500 nm diameter”, not “dimeter”.

9. Page 4, line 21. The figure number in “Fig. A-E” is missing.

10. Fig. 4. Subplots F and L are not referenced in the main text.

11. Page 4, lines 33-34. In the sentence “To understand the τ_O (unfolded dwell time) and the τ_C (folded dwell time) depends on RNA concentration and enzyme concentration”, “depends” should be corrected with “dependance”.

12. Fig. S7-F. The k_{cat} value in the plot (90 s⁻¹) does not match the value reported in the caption (98 s⁻¹). This is indeed much faster than the reported value in the literature. Do the authors know if RNase H behaves as an endonuclease or exonuclease based on the RNA binding mode in their system? According to Lee et al (Nucleic Acids Research, 2022, Vol.50, No.4), “RNase H acts as a processive exoribonuclease on the 3' DNA overhang side but as a distributive non-sequence-specific endonuclease on the 5' DNA overhang side of RNA:DNA hybrids or on blunt-ended hybrids”. The type of enzymatic activity exhibited by RNase H is expected to affect the kinetics as well.

13. Fig. S9 is not referenced in the main text.

14. Information regarding the sources of reagents and materials used in this study, including whether purchased from a vendor or synthesized, is not provided in the Supplementary Information file. More detailed information about the microscope setup (e.g., TIRF configuration, inverted microscope model, laser specifications, dichroic mirrors) should also be included as these details are highly relevant to the experiments conducted.

Reviewer #2

(Remarks to the Author)

Reviewer #3

(Remarks to the Author)

The authors developed an autonomous micron-scale molecular actuator based on a DNA origami structure, driven by RNA hydrolysis catalyzed by an enzyme RNase H in solution. By immobilizing one end of the actuator to a glass surface and attaching another end to a 500-nm polystyrene bead probing motion, the authors successfully monitored repetitive conformational changes between folded and unfolded states. The authors showed that the dwell times of these states depend on temperature and RNA and RNase H concentrations, and the dwell times as short as ~10 s have been achieved at high RNA and RNase H concentrations. Interestingly, the dwell time for the unfolded state depended on both RNA and RNase H concentrations, while the dwell time for the folded state only depended on RNase H concentration. Then, to reproduce folded and unfolded dwell times dependent on RNA and RNase H concentrations, two kinetic models were developed. I think that this molecular machine is well designed, and its autonomous conformational changes are verified properly. This is an excellent example of a micron-scale molecular machine which operates autonomously in the presence of a chemical fuel. On the other hand, the word "motor" is misleading because this molecular machine is an actuator rather than a motor. In addition, the materials and methods are not described sufficiently to reproduce the research, and there is significant duplication of the content between the main text and supplementary materials. Furthermore, "a double-exponential model" used for the fitting of the cumulative distribution of the dwell time is not clear at all. Therefore, extensive revisions are required before publication.

Major points

1. The authors used the word "motor" to describe their molecular machine, including the title. However, in my opinion, "actuator" is more appropriate than "motor", because it shows switch-like conformational changes between two states (folded and unfolded). It should be noted that biological molecular motors show linear or rotary motions, not switch-like motions demonstrated in this study.
2. The authors should clarify the sources of the materials (DNA, RNA, RNase H, polystyrene bead, and other chemical reagents and proteins) in the materials and methods. For example, although RNase H is one of the most important materials used in this study, no description is provided regarding its source, although I guess it is commercially available *E. coli* RNase H considering the reference 35 (main text, page 4, line 143). If the authors used commercially available products, the manufacturers, product names, and product numbers should be provided for all materials.
3. The authors should clarify the details of the instruments (for examples, manufacturers, product names, and product numbers of a fluorescence microscope and a high-speed camera) used in this study in the materials and methods.
4. There is significant duplication of the content between the methods in the main text and the materials and methods in the supplementary materials. I think the methods in the main text can be omitted, and the descriptions about the results, discussion, and conclusion can be expanded and deepened.
5. The authors should clarify the time resolutions of the particle tracking experiments using high-speed camera in the figure legends and the materials and methods. Although I found that 20-second or 2-second running averaged time trajectories was used for the analysis of the dwell time (legends of Figure 3 and 5, respectively), I could not find the time resolutions of the raw trajectories.
6. The authors should clarify the experimental temperatures for the data shown in Figures 2, 4, and 5. Please describe them in the figure legends and the material and methods.
7. It is not easy to understand the correspondence between the schematic illustration in Fig. S11 and the actual sequences of DNA sticky ends and hinge springs in Table S3. Please clarify the correspondence by adding the specific names of the sequences in Fig. S11 and describing which ones correspond to DNA sticky ends and hinge springs in Table S3.
8. Fig. S4, Table S1, and Table S3. It is not clear what kinds of the sequences are used as the DNA sticky ends and the hinge springs for RNA linkers other than RNA linker-32SE and RNA linker-60SE. Please describe all the sequences in Table S3. In my understanding, the DNA sticky ends of 6HB-32SE-008 and 6HB-60SE-008, and the hinge springs of 6HB-Hinge-1T-32SE-174 and 6HB-Hinge-1T-60SE-174 were used for RNA linker-32SE and RNA linker-60SE, respectively.
9. For the fitting of the cumulative distributions of dwell times to extract the time constants for folded and unfolded states, "a double-exponential model" was used (Figures 2C, 3B-G, 4A-K, and 5C). However, there is no explanation of the reasons or basis for using a double-exponential model. Does this mean that a single-exponential model cannot fit the experimental data well? Furthermore, if the double-exponential model is used for the fitting, two different values of the time constants will be obtained for each cumulative distribution, and it is not clear why only one time constant is described for each distribution.

The authors should clarify these points.

10. Related to the comment #9, I found sharp increases in the cumulative probability from 0 to 0.2~0.4 in the beginning of some cumulative distributions (for examples, Figures 2C, 3B, 3C, 4A-D, 4G-I). Are these sharp increases caused by the 20-second running averaging of the trajectories? Is this the reason why a double-exponential model was used? Please clarify.

11. Related to the comment #9, in Fig. S7, the fittings shown in dashed red lines are single-exponential, based on the simulation model described in Fig. S6 and the supplementary text (supplementary materials, page 3, line 111). What is the rationale for using a single-exponential model here, while experimental cumulative distributions were fitted with a double-exponential model in the main text?

Minor points

1. The figures should be numbered in the order of appearance. For examples, Fig. S11 (first appears in main text, page 2, line 65) should be numbered as Fig. S2, and Fig. S2 (first appears in main text, page 6, line 259) should be numbered as Fig. S11.
2. Main text, page 3, line 92. "selectivity, ," should be ""selectivity," (remove one comma).
3. Main text, page 5, line 201. "about10 seconds(Fig. 5)" should be "about 10 seconds (Fig. 5)".
4. Main text, page 13, Figure 3. There is no panel "D" in this figure. The panels should be A to G, not A to H.
5. Main text, page 15, line 495. " μM)and" should be " μM) and".
6. Supplementary materials, page 4, line 131. The "reported" value of the k_{cat} ($\sim 0.1 \text{ s}^{-1}$) for RNase H is different from that ($\sim 1 \text{ s}^{-1}$) described in the main text (page 4, line 173). I think the later value ($\sim 1 \text{ s}^{-1}$) is correct.
7. Supplementary materials, page 12, Fig. S7. The estimated value of k_{cat} (90 s^{-1}) described in the panel F is different from that (98 s^{-1}) described in the figure legend and the supplementary text (supplementary materials, page 4, line 129).

Version 1:

Reviewer comments:

Reviewer #1

(Remarks to the Author)

General remarks:

The authors have carefully addressed my previous comments. Notably, the term "motor" has been replaced with "engine", which more appropriately characterizes the type of motion achieved by the hinge machine described in the manuscript. The new supplementary microscopy videos are great at elucidating the particle dynamics and provide compelling evidence of the DNA engine diffusing around two separate positions. Furthermore, newly added references help place this work within the context of existing studies on RNase H-driven DNA machines, thus clarifying the novelty of the present contribution. The newly added control experiments with a scrambled RNA linker also strengthen the authors' mechanistic claims by corroborating that folding events are driven by specific hybridization interactions.

The revised manuscript is close to being suitable for publication, provided that a few remaining concerns and minor issues are addressed.

Major concerns:

1. The cumulative distribution functions of dwell times τ_{O} in the unfolded state in Fig. S11 (previously Fig. S9) are not always well fitted as some red dashed lines do not adequately overlap with the experimental data (blue solid). To strengthen the analysis, the authors should include a goodness-of-fit test to evaluate how well the kinetic model agrees with observed data.

Minor issues:

1. In Supplementary Data 1-3, the timelapse videos recorded using fluorescence microscopy should display a scale bar. In Supplementary Data 3, the colloidal particle appears larger than in previous videos, suggesting that a different magnification was used. It would be helpful for the authors to clarify or standardize the imaging parameters.
2. The caption for Table S3 should be updated as the origami design created in caDNAno has been moved from Fig. S11 to S2.
3. The caption for Fig. 4L is incorrect. The graph plots τ_{O} and τ_{C} against enzyme concentration, not RNA concentration.
4. In the revised version of the Supplementary Information, Supplementary Text 2 ("Simulation of the τ_{C} of the RNA-fueled engine") and 3 ("Simulation of the τ_{O} of the RNA-fueled engine") still refer to the previous figure numbering. These should be updated to match the current figure order.
 - a) The three-state model of RNA hydrolysis by RNase H enzyme is now illustrated in Fig. S8 (not S6).
 - b) Cumulative distribution functions (CDFs) of the folded-state dwell times (τ_{C}) at various enzyme concentrations are in Fig.

S9 (not S7).

c) The kinetic pathway model describing the dwell time in the unfolded state is in Fig. S10 (not S8).

5. Similarly, the caption for Fig. S12 should be updated to reference the correct figure numbers.

Reviewer #2

(Remarks to the Author)

Reviewer #3

(Remarks to the Author)

I think the authors adequately addressed my comments and those of other reviewers. I support publication, but request the following minor additional revisions for better clarity and correction of errors.

1. Page 3, Lines 132 and 134. "double -exponential" should be "double-exponential".
2. Page 3, Line 133. "exponential(Fig. S6" should be "exponential (Fig. S6".
3. Page 4, Lines 135 and 152. "the dwell times" should be "the mean dwell times" for better clarity.
4. Page 4, Line 142. Please add the following sentence after "(Fig. S7B)." to explain "a scrambled RNA linker" for better clarity.
"The scrambled linker preserves the length but was sequence-shuffled to eliminate complementarity to both sticky ssDNA ends (Table S1)."
5. Page 5, Line 181. "hybridization(Fig. 4L)." should be "hybridization (Fig. 4L)."
6. Page 5, Line 220. "about10" should be "about 10".
7. Page 6, Line 266, and Page 7, Line 272. "Information.." should be "Information."
8. Supplementary Materials, Page 5, Line 154. "AICdoubleand" should be "AICdouble and".
9. Supplementary Materials, Page 5, Line 167. "{ti}with" should be "{ti} with".
10. Supplementary Materials, Page 8, Fig. S2. Please enlarge the sequence names in this figure for better clarity.
11. Supplementary Materials, Page 10, Line 254. "Table S3.." should be "Table S3."
12. Supplementary Materials, Page 12, Line 267. "CDF(*t*)from" should be "CDF(*t*) from".
13. Supplementary Materials, Page 22, Line 351. "Fig. S11" should be "Fig. S2".

Version 2:

Reviewer comments:

Reviewer #1

(Remarks to the Author)

The authors have now addressed the comments and critiques.

Reviewer #2

(Remarks to the Author)

REPLY TO REVIEWERS

We thank the reviewers for their time and thoughtful evaluation of our work. We appreciate the constructive feedback and have addressed every point in the point-by-point responses below. Reviewer comments appear in plain text, author responses are shown in light blue, and changes to the manuscript are highlighted.

Reviewer #1 (Remarks to the Author):

General remarks:

This article by Wang & Chaikin et al reports the design of a DNA origami hinge machine that switches between folded and unfolded conformations upon RNA hybridization and enzymatic cleavage by RNase H respectively, driving micrometer-scale movements of a 500 nm-diameter particle. The hinge machine is comprised of four 6-helix bundle (6HB) units assembled into two arms: a short, immobilized arm and a longer, semi-rigid moving arm. The two arms are connected through a flexible hinge spring composed of two single-stranded DNA domains, and both present sticky ssDNA protrusions close to the hinge. Introduction of an RNA linker that bridges the sticky DNA ends induces hinge closing, whereas RNase H-mediated hydrolysis of the bound RNA releases the stored elastic energy, restoring the origami structure to its unfolded conformation.

In their previous work, Wang and Chaikin et al demonstrated the ability to control the folding and unfolding transitions of an analogous DNA origami construct using a microheater to change the local temperature (Science Advances, 2025). The main difference in the molecular hinge design was that the sticky DNA protrusions on opposing arms could directly hybridize. At lower temperatures, hybridization stabilized the folded state, while higher temperatures induced duplex melting, favoring the open conformation. Each transition was precisely controlled by external input as applying current through the heater enabled rapid equilibration to the programmed temperature within milliseconds.

The new proposed work is an interesting implementation of the RNase H-powered consumption of RNA fuel molecules for driving autonomous motion and manipulation of microscale or nanoscale objects. Unlike the previously reported heat-activated hinge machine, this RNA-fueled machine is fully autonomous. This property also sets this work apart from the class of photoactivated DNA switches, which lack both processivity and autonomy (Willner et al, Angew. Chem. Int. Ed., 2017; Kuzyk et al, Nature Communications, 2016). While the design of a lever arm using DNA origami technology is not itself novel, this work presents some clear scientific merit.

Furthermore, the PAGE assay included to determine the specific activity of RNase H on the

selected RNA sequence is valuable as the enzyme is known to exhibit sequence preferences as well as non-specific cleavage with certain sequences (Kielinski et al, Nucleic Acids Research, 2017, Vol.45, No.22). Hence, validating that RNase H enzyme selectively cleaves RNA substrate in DNA:RNA hybrids employed in the system, while leaving ssDNA and ssRNA intact, was an important and well-justified step. The mechanistic insights and kinetic analysis are also thorough and add value to the paper.

That said, I have a few concerns that require the active consideration of the authors. My comments are listed below:

Major concerns:

1. The authors claim that the hinge machine is an RNA-fueled “motor”. However, the DNA machine described in this work behaves more like a molecular switch that transitions between two states, folded and unfolded, rather than a processive motor that continuously converts chemical energy into net mechanical work. While the attached particle should move across micrometer distances, we only observe the endpoints, reinforcing the interpretation of a binary switch.

Based on the extent of motility and type of mechanical movement, the hinge machine reported in this study is more akin to a nano-engine, similar to the one developed by Centola & Famulok et al (Nature Nanotech., 2024) that generates a rhythmic pulsating motion powered by T7 RNA polymerase. The distinction between DNA switches and DNA motors is a critical one, and these terms should not be used interchangeably.

If a switching mechanism, which on its own cannot generate net mechanical work, is incorporated into a motor system that produces net mechanical work over a cycle in the form of translocation along a linear track or across a 2D surface, it would better qualify as a motor. An example is the bar-hinge motor reported by Small et al (New Journal of Physics, 2019). However, the RNase H-activated hinge machine described in this study lacks this property.

We thank the reviewer for this careful and constructive comment. We agree that, in its current implementation, our RNA-fueled engine operates as a binary switching system. In accordance with the reviewer’s recommendation, we have revised the manuscript to avoid any overstatement. We have replaced the term “*motor*” with “*engine*” across the entire manuscript, including the Title, Abstract, Main Text, and Figure captions. We believe these changes address the reviewer’s concern and more accurately reflect the demonstrated capabilities and limitations of our device.

2. Building upon my previous comment, I strongly suggest including fluorescence microscopy snapshots of single particles over time. Real-time fluorescence tracking data revealing particles that dwell in intermediate positions could provide evidence for motor-like behavior. For example, the electric field-driven DNA rotary motor reported by Kopperger & Simmel et al (Science, 2018) was shown at different positions along the circular trajectory in snapshots taken using fluorescence microscopy.

We appreciate this suggestion. In our setup, the readout is the relative position of a fluorescent bead (tracked against a fixed reference), and single-frame snapshots offer limited contrast and temporal fidelity for resolving brief intermediate dwells. To provide a clearer and more faithful visualization, we now include three short movies: Supplementary Data 1 (unfolded control), Supplementary Data 2 (folded control; RNA only), and Supplementary Data 3 (autonomous switching with RNA + RNase H). These recordings directly show the particle dynamics in each state and during switching, which are more informative than isolated snapshots. The corresponding description and cross-references have been added to the Main Text and Supplementary Information.

3. A search of the literature quickly reveals that there is important literature on “RNA-Fueled Micro-Motors” that were not cited or mentioned in this report. This needs to be cited: 1) Yehl et al. 2016 Nature Nanotechnology (this paper shows the first RNA fueled micromotor). 2) Bazrafshan et al. 2020 Angewandte Chemie (this paper shows the first RNA fueled origami motor).

We thank the reviewer for highlighting these key literature. In the revised manuscript we now explicitly cite Yehl *et al.* (Nat. Nanotechnol., 2016) and Bazrafshan *et al.* (Angew. Chem. Int. Ed., 2020), and we have added a brief contextual paragraph clarifying that Yehl *et al.* established RNA as a fuel for autonomous motion in nucleic-acid systems, while Bazrafshan *et al.* demonstrated RNA-fueled actuation on a DNA-origami motor architecture. We further state that, consistent with the reviewer’s earlier comment and our revised terminology, our device is framed as an ‘RNA-fueled micro-engine’. These additions ensure that the manuscript accurately reflects prior art and positions our contribution within the established literature on RNA-fueled engine.

4. In Fig. 2A, how do the authors ensure a preferential orientation of hinge machines in their open conformation? The particle distribution appears skewed, with a higher density in the bottom-right region of the field of view. How is the orientation of the origami arms controlled during surface immobilization? One would expect the origami structures to deposit randomly, both in position and orientation, resulting in a more homogeneous distribution across the

surface.

We do not impose a global orientation during deposition; both the positions and azimuthal orientations of individual devices are random prior to tethering. Orientation becomes fixed only after multipoint anchoring to the surface (Fig. S1): the 6HB-1 arm carries two adjacent helices bearing a line of capture sticky ends that hybridize to complementary strands on the substrate, thereby defining a unique polarity of the device and preventing post-binding rotation. The 6HB-1 and 6HB-2 is connected through a hinge spring composed of two short ssDNA strands (Fig. S1 and Fig. S2B), which biases the direction of folding/unfolding relative to the anchored arm, but does not enforce a common compass direction across the field of view. The mild skew in particle density seen in Fig. 2A reflects local heterogeneity in capture-strand density within that particular field, rather than a preferred global orientation of the hinges.

5. FITC (Fluorescein isothiocyanate) is used as fluorescent label on the polystyrene colloidal particles for tracking purposes. FITC is widely used for applications that do not require high sensitivity and/or long exposure because it is less bright than other green-fluorescing dyes and it is prone to rapid photobleaching. Photobleaching occurs when the fluorescent molecule is irreversibly degraded by exposure to high-intensity light and loses its ability to fluoresce. There are several alternative dyes with significant greater brightness and photostability, such as Alexa fluor 488. In this study, the experimental design does not impose restrictions related to orthogonal fluorophore requirements, so alternative dyes that are excited or emit at different wavelengths (e.g., ATTO 565, ATTO 647N) could be readily employed.

We agree with the reviewer that FITC is less photostable and more susceptible to bleaching than several modern dyes. In the present experiments we mitigated photobleaching by using low excitation intensities; because the tracer is a 500-nm bead densely labeled with FITC, the resulting signal-to-noise ratio remained sufficient for reliable tracking over our acquisition windows. We will adopt more photostable fluorophores (e.g., Alexa Fluor 488, ATTO 565, ATTO 647N) to further improve brightness and stability.

Do the authors employ any software, such as ImageJ Fiji, to correct for the diminishing signal caused by photobleaching? It is somewhat surprising that the FITC signal remains sufficiently bright given the extended imaging duration (over 30 minutes) and the high acquisition frequency reported in this study.

The reviewer is correct: as expected, prolonged imaging led to a gradual intensity decay. We mitigated this in three ways: (1) low excitation intensity; (2) dense labeling—the 500-nm beads are densely labeled, so the signal remained sufficient for reliable localization even as intensity declined; and (3) Fiji/ImageJ contrast adjustment—image stacks were pre-processed in Fiji/ImageJ (contrast adjusted) to facilitate tracking with the Trackpy package. We have added

these details to Methods (Data Recording and Particle Tracking) in the Supplementary Information.

Given the size of the colloidal particle (500 nm diameter), label-free approaches like reflection interference contrast microscopy (RICM) and dark-field scattering microscopy should be effective for single-particle tracking. RICM was used to track 50-nm gold nanoparticle motors (Bazrafshan et al, ACS Nano, 2021), and dark-field scattering microscopy can track 100-nm gold nanoparticles with 1-ms time resolution and 1-nm localization precision (Harashima et al, Nature Communications, 2025).

We agree that label-free modalities such as RICM and dark-field scattering are well suited to track the engine's behavior and could improve temporal precision and state separability, particularly when combined with shorter lever arms or smaller metallic tracers. In the present work, given our available instrumentation, we tracked densely labeled 500-nm beads and used position localization against a fixed reference, which provided sufficient signal-to-noise for the reported analyses. We have added a statement in the main text acknowledging RICM and dark-field as compelling alternatives and outlining their anticipated benefits for future iterations of the platform.

6. Although the authors provide convincing data showing that switching kinetics can be tuned by changing key parameters (temperature, RNA concentration or enzyme concentration), the underlying switching behavior may merely reflect the confined diffusion of the attached particle. As a control, time series of particle relative position of individual origami-particle complexes in the absence of both RNA and RNase H should be included. Such data would strengthen the mechanistic interpretation by demonstrating that the hinge remains in the open conformation for the full duration of the timelapse experiment without open/close shifts. It is also possible that the release of elastic energy following the first RNase H cleavage event might be sufficient to generate an oscillatory motion of the lever arm with the colloidal particle.

We appreciate the reviewer's request for diffusion controls and have added new measurements that directly address this point. In the absence of both RNA and RNase H, single-particle time-lapse traces (now Supplementary Fig. S7A) show that the device remains in the unfolded conformation for the entire recording (≥ 30 min), exhibiting only thermal fluctuations around a single mean value without transitions. Upon subsequent addition of RNA (red segment), the signal enters the folded plateau, confirming that the prior fluctuations are not attributable to latent switching. Regarding the possibility of oscillations driven by elastic energy release after the first RNase H cleavage, we do not observe sustained or damped periodic signals; it remains possible that any small-amplitude oscillations are below our detection limit and masked by thermal noise, and the running-average preprocessing would further attenuate such weak periodic components.

7. A control experiment using a scrambled RNA linker that fails to bridge the sticky ssDNA ends should be included. Since no strategy to visualize RNA hydrolysis is being used, this control would help confirm that hinge closing arises specifically from RNA hybridization rather than from non-specific interactions.

We appreciate this valuable suggestion. We have added a scrambled-RNA control. The scrambled linker preserves the length but was sequence-shuffled to eliminate complementarity to both sticky ssDNA ends. In the presence of scrambled RNA—both without (green segment) and with (gray segment) RNase H—the hinge remained in the unfolded state, with no transitions detected (Supplementary Fig. S7B). By contrast, the complement RNA linker drove entry into the folded plateau (red segment). Together, these results show that hinge folding arises from sequence-specific RNA hybridization rather than nonspecific interactions. We have added these data and sequence details to the Supplementary Information (Table S1 and Fig. S7B).

Minor issues:

1. The DNA origami is most likely designed using caDNAo, as also suggested by the cropped images in Fig. S11. However, neither the main article nor the Supporting Information cite this widely used, open-source software package (Douglas & Shih et al., Rapid prototyping of 3D DNA-origami shapes with caDNAo, *Nucleic Acids Research*, 2009).

Thank you for the suggestion. We have revised the Supporting Information to explicitly state that caDNAo was used for design and have added the appropriate citation to Douglas et al., *Nucleic Acids Research* (2009).

2. Supplementary figures are not cited in a consecutive order in the main text. For example, S11 is referenced right after S1 (page 2, line 24), and S2 does not appear before S3. I suggest rearranging the SI figures according to the first time they are mentioned in the main text.

We have reordered and renumbered all Supplementary Figures according to their first mention in the main text and updated the in-text citations and captions to make them easier to locate.

3. Suggest adding the fragment sizes in bp for the bands in the DNA ladder in Fig. S3.

We have added the DNA ladder fragment sizes (bp) to Fig. S3 and updated the figure accordingly. The ladder specifications are also provided in the Supplementary Information.

4. The screening of RNA linkers of various length via non-denaturing agarose gel electrophoresis in Figure S4 indicates that all RNA linkers from 28 nt to 58 nt produce dual

populations of folded and unfolded 6HB dimers, as evidenced by two distinct bands under each condition. This suggests that some 6HB dimers fail to fold into the closed conformation in the presence of the RNA linker. What is the ratio of origami to RNA? How long are they incubated together before gel analysis?

Thank you for the thoughtful comment. As shown in Fig. S4, a subset of 6HB dimers remains unfolded in the presence of an RNA linker, consistent with the off-pathway scenario in Fig. S8. We attribute the unfolded band to a mis-bridged state in which two RNA linkers independently occupy the two sticky ends. For the gel in Fig. S4, samples were prepared at a 1:10 DNA-origami : RNA-linker molar ratio and incubated for 10 min at room temperature prior to native agarose analysis. We have added these ratio and incubation details to the Supplementary Information.

5. Could shorter RNA linkers with higher GC content lead to faster binding/folding rate while maintaining duplex stability?

Thank you for the insightful question. In our design we selected the shortest RNA linkers to maximize RNase H-mediated cleavage and thereby accelerate the unfolding step. Regarding the folding step, shorter linkers with higher GC content are indeed expected to shorten the folding time, provided duplex stability and geometric reach are maintained. These gains must be balanced against the risk that very GC-rich sequences introduce secondary structure or off-target interactions. Within our kinetic framework (Fig. S10), the cycle time reflects both RNA binding (folding) and RNase H turnover (reset), so judicious shortening can benefit both steps up to the point where span, specificity, or cleavage efficiency become limiting.

6. The word “colloid” (not “collid”) is spelled incorrectly in Fig. S1.

Thank you for catching this. We have corrected the spelling from “collid” to “colloid” in Fig. S1 and updated the figure accordingly.

7. Double commas on page 3, line 5.

Thank you for noting this. The extra comma on page 3, line 5 has been removed, and the text has been updated accordingly.

8. There is a typo on page 3, line 15. It should be “500 nm diameter”, not “dimeter”.

Thank you for pointing this out. We’ve corrected “dimeter” to “diameter” (now “500 nm diameter”), and updated accordingly.

9. Page 4, line 21. The figure number in “Fig. A-E” is missing.

Thank you for pointing that out. We have corrected the reference to include the full figure number—now properly cited as “Fig. 4A–E”.

10. Fig. 4. Subplots F and L are not referenced in the main text.

Thank you for the careful observation. We have revised the main text to include references to subplots Fig. 4F and Fig. 4L, ensuring all figure panels are properly cited and discussed.

11. Page 4, lines 33-34. In the sentence “To understand the τ_O (unfolded dwell time) and the τ_C (folded dwell time) depends on RNA concentration and enzyme concentration”, “depends” should be corrected with “dependance”.

Thank you for pointing this out. We have corrected the sentence to read: “To understand the τ_O (unfolded dwell time) and τ_C (folded dwell time) dependence on RNA concentration and enzyme concentration...”

12. Fig. S7-F. The k_{cat} value in the plot (90 s^{-1}) does not match the value reported in the caption (98 s^{-1}). This is indeed much faster than the reported value in the literature.

Thank you for the careful observation. We have corrected the inconsistency in caption—the plotted value has been updated to (98 s^{-1}), and we have double-checked all corresponding text for accuracy.

Do the authors know if RNase H behaves as an endonuclease or exonuclease based on the RNA binding mode in their system? According to Lee et al (Nucleic Acids Research, 2022, Vol.50, No.4), “RNase H acts as a processive exoribonuclease on the 3' DNA overhang side but as a distributive non-sequence-specific endonuclease on the 5' DNA overhang side of RNA:DNA hybrids or on blunt-ended hybrids”. The type of enzymatic activity exhibited by RNase H is expected to affect the kinetics as well.

We appreciate the reviewer’s insightful comment. As highlighted by Lee et al. (Nucleic Acids Research, 2022), RNase H exhibits mode-dependent activity on RNA–DNA hybrids. It behaves as a processive exonuclease when a 3' DNA overhang flanks the hybrid, and as a distributive, non–sequence-specific endonuclease when a 5' DNA overhang or a blunt end is presented. In our design the RNA recognition site is fully paired within the hybrid and we did not intentionally introduce a 3' DNA overhang at the cleavage boundary. By geometry our substrates therefore fall into the 5'-overhang or blunt category, so RNase H is expected to act as a distributive endonuclease rather than a processive exonuclease. Our current kinetic framework does not directly resolve the enzymatic mode. The parameters we report should be interpreted as apparent or effective quantities. We cannot determine from these data whether the mode materially alters the observed timescales. In principle, under enzyme-excess conditions a distributive mode could accelerate the apparent kinetics by permitting independent enzyme engagements, but this remains a hypothesis in our configuration.

13. Fig. S9 is not referenced in the main text.

Thank you for your comment. We have updated the main text to include a reference to Fig. S9 (now Fig. S11), ensuring all supplementary figures are properly cited.

14. Information regarding the sources of reagents and materials used in this study, including whether purchased from a vendor or synthesized, is not provided in the Supplementary Information file. More detailed information about the microscope setup (e.g., TIRF configuration, inverted microscope model, laser specifications, dichroic mirrors) should also be included as these details are highly relevant to the experiments conducted.

Thank you for your comment. We have added a dedicated “Reagents and materials” subsection to the Supplementary Information listing the source for every component, together with catalog numbers where applicable. We also expanded “Data recording and particle tracking” to detail the optical configuration: Nikon Eclipse Ti inverted microscope (no TIRF used), 100× oil-immersion Plan Apo objective, Andor Zyla sCMOS detector, SPECTRA Light Engine (Lumencor), Chroma 49002 (ET-EGFP/FITC/Cy2) filter set. These additions are now included in the Supplementary Information.

Reviewer #2 (Remarks to the Author):

We are grateful for your contribution and for participating in the Nature Communications co-review initiative that recognizes Early Career Researchers.

Reviewer #3 (Remarks to the Author):

The authors developed an autonomous micron-scale molecular actuator based on a DNA origami structure, driven by RNA hydrolysis catalyzed by an enzyme RNase H in solution. By immobilizing one end of the actuator to a glass surface and attaching another end to a 500-nm polystyrene bead probing motion, the authors successfully monitored repetitive conformational

changes between folded and unfolded states. The authors showed that the dwell times of these states depend on temperature and RNA and RNase H concentrations, and the dwell times as short as ~10 s have been achieved at high RNA and RNase H concentrations. Interestingly, the dwell time for the unfolded state depended on both RNA and RNase H concentrations, while the dwell time for the folded state only depended on RNase H concentration. Then, to reproduce folded and unfolded dwell times dependent on RNA and RNase H concentrations, two kinetic models were developed. I think that this molecular machine is well designed, and its autonomous conformational changes are verified properly. This is an excellent example of a micron-scale molecular machine which operates autonomously in the presence of a chemical fuel. On the other hand, the word “motor” is misleading because this molecular machine is an actuator rather than a motor. In addition, the materials and methods are not described sufficiently to reproduce the research, and there is significant duplication of the content between the main text and supplementary materials. Furthermore, “a double-exponential model” used for the fitting of the cumulative distribution of the dwell time is not clear at all. Therefore, extensive revisions are required before publication.

Major points

1. The authors used the word “motor” to describe their molecular machine, including the title. However, in my opinion, “actuator” is more appropriate than “motor”, because it shows switch-like conformational changes between two states (folded and unfolded). It should be noted that biological molecular motors show linear or rotary motions, not switch-like motions demonstrated in this study.

We appreciate the reviewer’s clarification. In line with this recommendation, and a similar suggestion by another reviewer, we now describe the device as an RNA-fueled engine rather than a motor throughout the manuscript, including the title, abstract, main text, and figure captions. These revisions align the terminology with the behavior we report and address the reviewer’s concern regarding overstatement.

2. The authors should clarify the sources of the materials (DNA, RNA, RNase H, polystyrene bead, and other chemical reagents and proteins) in the materials and methods. For example, although RNase H is one of the most important materials used in this study, no description is provided regarding its source, although I guess it is commercially available E. coli RNase H considering the reference 35 (main text, page 4, line 143). If the authors used commercially available products, the manufacturers, product names, and product numbers should be provided

for all materials.

We thank the reviewer for this helpful suggestion. We have added a “Reagents and materials” subsection specifying the vendor, product name, and catalog number for all commercially sourced items, including DNA/RNA oligonucleotides (Integrated DNA Technologies), the M13mp18 scaffold (Tilbit Nano systems GmbH, type p7249), E. coli RNase H (NEB, M0297L), streptavidin and biotin-BSA (Sigma-Aldrich), polystyrene microspheres (Bangs Laboratories), and all other chemicals (Sigma-Aldrich).

3. The authors should clarify the details of the instruments (for examples, manufacturers, product names, and product numbers of a fluorescence microscope and a high-speed camera) used in this study in the materials and methods.

We appreciate the reviewer’s comment. We expanded “Data recording and particle tracking” to detail the optical configuration: Nikon Eclipse Ti inverted microscope (no TIRF used), 100× oil-immersion Plan Apo objective, Andor Zyla sCMOS detector, SPECTRA Light Engine (Lumencor), Chroma 49002 (ET-EGFP/FITC/Cy2) filter set. These additions are now included in the Supplementary Information.

4. There is significant duplication of the content between the methods in the main text and the materials and methods in the supplementary materials. I think the methods in the main text can be omitted, and the descriptions about the results, discussion, and conclusion can be expanded and deepened.

Thank you for the suggestion. We have revised the manuscript accordingly: the detailed experimental procedures have been removed from the main-text Methods and consolidated in the Supplementary Information, where they are organized by assay and figure. The main text now retains only a brief methodological overview.

5. The authors should clarify the time resolutions of the particle tracking experiments using high-speed camera in the figure legends and the materials and methods. Although I found that 20-second or 2-second running averaged time trajectories was used for the analysis of the dwell time (legends of Figure 3 and 5, respectively), I could not find the time resolutions of the raw trajectories.

Thank you for the comment. The raw particle trajectories were acquired at 20 FPS for all experiments except Fig. 5, which used a high-speed rate of 200 fps to resolve short dwells. We have added these acquisition rates to the figure legends.

6. The authors should clarify the experimental temperatures for the data shown in Figures 2, 4, and 5. Please describe them in the figure legends and the material and methods.

Thank you for the comment. The raw particle trajectories were acquired at 37 °C except Fig. 3, which used different temperature to investigate the temperature-dependent kinetics of the RNA-fueled engine. We have added these temperatures to the figure legends and to Methods (Data recording and particle tracking).

7. It is not easy to understand the correspondence between the schematic illustration in Fig. S11 and the actual sequences of DNA sticky ends and hinge springs in Table S3. Please clarify the correspondence by adding the specific names of the sequences in Fig. S11 and describing which ones correspond to DNA sticky ends and hinge springs in Table S3.

We appreciate the suggestion. We have updated Fig. S11 (now Fig. S2) and Fig. S4 by adding the specific sequence names that exactly match Table S3. We also revised the figure caption to explicitly state this correspondence.

8. Fig. S4, Table S1, and Table S3. It is not clear what kinds of the sequences are used as the DNA sticky ends and the hinge springs for RNA linkers other than RNA linker-32SE and RNA linker-60SE. Please describe all the sequences in Table S3. In my understanding, the DNA sticky ends of 6HB-32SE-008 and 6HB-60SE-008, and the hinge springs of 6HB-Hinge-1T-32SE-174 and 6HB-Hinge-1T-60SE-174 were used for RNA linker-32SE and RNA linker-60SE, respectively.

Thank you for the helpful comment. We have expanded the caption of Fig. S2 and Fig. S4 to indicate which RNA-linker length each pair was used with, and we redrew Fig. S4 with a revised caption to display these pairings explicitly. Main experiments used (all figures except Fig. S4C–D): RNA linker-32SE and Sticky ends: 6HB-32SE-008, 6HB-Hinge-1T-32SE-174. Length-scan assay (Fig. S4C–D): to compare different RNA-linker lengths while keeping DNA constant, we used the same long sticky-end pair for all lanes—Sticky ends: 6HB-60SE-008, 6HB-Hinge-1T-60SE-174—with RNA linkers of 12, 16, 20, 24, 28, 32, 36, 40, 44, 48, 52, 56, 58, and 60 nt. Controls used DNA-only folding (6HB-Control-Fold-008 with 6HB-Hinge-3T-Control-Fold-174) and a no-RNA (unlinked) condition. These mappings are now shown by the labels in Fig. S2B and Fig. S4, ensuring one-to-one correspondence between figure labels and Table S3 sequence names.

9. For the fitting of the cumulative distributions of dwell times to extract the time constants for

folded and unfolded states, “a double-exponential model” was used (Figures 2C, 3B-G, 4A-K, and 5C). However, there is no explanation of the reasons or basis for using a double-exponential model. Does this mean that a single-exponential model cannot fit the experimental data well? Furthermore, if the double-exponential model is used for the fitting, two different values of the time constants will be obtained for each cumulative distribution, and it is not clear why only one time constant is described for each distribution. The authors should clarify these points.

We appreciate the reviewer’s thoughtful question. We used a two-exponential survival model because the dwell-time distributions are not governed by a single Poisson process. Each macrostate (unfolded/folded) contains multiple microscopic substates. Model selection based on maximum-likelihood fits strongly favored a two-timescale description for both macrostates. Specifically, compared with a single-exponential survival we obtained Δ AIC (Akaike information criterion)= 771.4 and 713.5, and Δ BIC (Bayesian information criterion)= 759.7 and 701.7 (two datasets, unfolded/folded respectively). A likelihood-ratio test (LRT) with a parametric bootstrap under H_0 (single-exponential) gave $p < 2.5 \times 10^{-3}$ in both cases ($B = 400$) (Supplementary Text 1). Thus, the improvement provided by the two-exponential model is extremely unlikely if a single timescale were true.

Regarding parameter reporting, mixture components are partially covariant; therefore, rather than listing two separate time constants for each distribution, we summarize kinetics by the mean dwell time $\langle t \rangle = w \tau_f + (1-w) \tau_s$, where t is the dwell time, w is the mixing weight of the fast component ($0 < w < 1$), τ_f is the characteristic time of the fast subpopulation, and τ_s is the characteristic time of the slow subpopulation. This summary is robust and directly comparable across conditions. We have added these justifications to the and Supplementary text 1 (model forms, fitting, AIC/BIC, bootstrap LRT) and provide log-survival comparisons in Supplementary Fig. S6.

10. Related to the comment #9, I found sharp increases in the cumulative probability from 0 to 0.2~0.4 in the beginning of some cumulative distributions (for examples, Figures 2C, 3B, 3C, 4A-D, 4G-I). Are these sharp increases caused by the 20-second running averaging of the trajectories? Is this the reason why a double-exponential model was used? Please clarify.

We agree that some cumulative distributions show a sharp early rise. This feature is not produced by the 20 s running average. A running average is a low-pass filter that merges very brief recrossings into neighboring dwells and lengthens short events. It therefore suppresses, rather than creates, an excess of very short dwells. If anything, smoothing would make the early portion of the CDF rise more slowly, not more steeply.

The observed early rise reflects genuine heterogeneity in the kinetics. For the folded macrostate (τ_c) the timescale is governed by catalysis. A fast subpopulation cleaves promptly once

productively bound, effectively set by k_{cat} . A slow subpopulation is binding-limited or requires a conformational rearrangement, which lowers the productive bound fraction and lengthens the dwell. For the unfolded macrostate (τ_o) the system first captures RNA, then two competing routes operate on the DNA~RNA state. Successful-folding attempts give short dwells, repeated “bind–cleave–reset” cycles give long dwells. Pooling these two classes yields the observed early rise and tail.

We adopted a double-exponential model because it captures both the early rise and the long tail and it is decisively favored by model selection on the raw dwell times. As detailed in our response to Comment 9, information criteria and a bootstrap likelihood-ratio test strongly support a two-timescale description over a single exponential. To avoid over-interpreting individual mixture components, we report the mean dwell time as the summary statistic $\langle t \rangle = w \tau_f + (1-w) \tau_s$ for cross-condition comparisons..

11. Related to the comment #9, in Fig. S7, the fittings shown in dashed red lines are single-exponential, based on the simulation model described in Fig. S6 and the supplementary text (supplementary materials, page 3, line 111). What is the rationale for using a single-exponential model here, while experimental cumulative distributions were fitted with a double-exponential model in the main text?

We appreciate the reviewer’s question. The dashed red curves in Fig. S7 are single-exponential-like because they summarize the output of our minimal three-state mass-action ODE ($\text{DNA} \cdot \text{RNA} \rightleftharpoons \text{DNA} \cdot \text{RNA} \cdot \text{Enzyme} \rightarrow \text{DNA} + \text{Enzyme}$) under a homogeneous-rate assumption. In this framework binding reaches a rapid pre-equilibrium and cleavage proceeds as one first-order step. In our model, we integrate the ODE to obtain the product fraction $\text{DNA}(t)$, which is the model-predicted CDF, and we globally fit all $[\text{Enzyme}]$ conditions simultaneously (shared k_a, k_d, k_{cat}). This ODE analysis benchmarks the homogeneous-rate limit and constrains the concentration dependence through the apparent parameters k_a, k_d, k_{cat} .

By contrast, the **experimental** single-molecule dwell-time CDFs display an early rise and a long tail, which indicate kinetic heterogeneity within the closed macrostate. For these experimental distributions we therefore fit a **two-exponential** survival and summarize each condition by the mean dwell time $\langle t \rangle = w \tau_f + (1 - w) \tau_s$. In short, Fig. S7 addresses the homogeneous ODE prediction and its $[\text{Enzyme}]$ dependence, while the main-text fits quantify the heterogeneity present in the experimental data.

Minor points

1. The figures should be numbered in the order of appearance. For examples, Fig. S11 (first

appears in main text, page 2, line 65) should be numbered as Fig. S2, and Fig. S2 (first appears in main text, page 6, line 259) should be numbered as Fig. S11.

We have reordered and renumbered all Supplementary Figures according to their first mention in the main text and updated the in-text citations and captions to make them easier to locate.

2. Main text, page 3, line 92. “selectivity, ,” should be ““selectivity,” (remove one comma).

Thank you for catching this typographical error. We have removed the extra comma.

3. Main text, page 5, line 201. “about10 seconds(Fig. 5)” should be “about 10 seconds (Fig. 5)”.

Thank you for catching this typographical spacing error. We have corrected in the main text.

4. Main text, page 13, Figure 3. There is no panel “D” in this figure. The panels should be A to G, not A to H.

Thank you for pointing this out. We have corrected the panel labeling to A–G in Figure 3, updated the figure file and caption accordingly, and revised all corresponding in-text references.

5. Main text, page 15, line 495. “ μM)and” should be “ μM) and”.

Thank you for catching this typographical spacing error. We have corrected “ μM)and” to “ μM) and” in the main text.

6. Supplementary materials, page 4, line 131. The “reported” value of the k_{cat} ($\sim 0.1 \text{ s}^{-1}$) for RNase H is different from that ($\sim 1 \text{ s}^{-1}$) described in the main text (page 4, line 173). I think the later value ($\sim 1 \text{ s}^{-1}$) is correct.

You are correct. This was a typographical error.

7. Supplementary materials, page 12, Fig. S7. The estimated value of k_{cat} (90 s^{-1}) described in the panel F is different from that (98 s^{-1}) described in the figure legend and the supplementary text (supplementary materials, page 4, line 129).

We appreciate the reviewer’s careful reading. The correct estimate is $k_{\text{cat}} = 98 \text{ s}^{-1}$ (from the global fit reported in Supplementary materials). We have corrected all typos.

REPLY TO REVIEWERS

We sincerely appreciate for the time and effort spent reviewing our manuscript. We have addressed every point in the point-by-point responses below. Reviewer comments appear in plain text, author responses are shown in light blue, and changes to the manuscript are highlighted.

Reviewer #1 (Remarks to the Author):

General remarks:

The authors have carefully addressed my previous comments. Notably, the term “motor” has been replaced with “engine”, which more appropriately characterizes the type of motion achieved by the hinge machine described in the manuscript. The new supplementary microscopy videos are great at elucidating the particle dynamics and provide compelling evidence of the DNA engine diffusing around two separate positions.

Furthermore, newly added references help place this work within the context of existing studies on RNase H-driven DNA machines, thus clarifying the novelty of the present contribution. The newly added control experiments with a scrambled RNA linker also strengthen the authors’ mechanistic claims by corroborating that folding events are driven by specific hybridization interactions.

The revised manuscript is close to being suitable for publication, provided that a few remaining concerns and minor issues are addressed.

Major concerns:

1. The cumulative distribution functions of dwell times τ_O in the unfolded state in Fig. S11 (previously Fig. S9) are not always well fitted as some red dashed lines do not adequately overlap with the experimental data (blue solid). To strengthen the analysis, the authors should include a goodness-of-fit test to evaluate how well the kinetic model agrees with observed data.

We appreciate the reviewer’s insightful suggestion regarding the quantitative evaluation of our kinetic model. We agree that a statistical assessment is essential to objectively validate the agreement between the experimental data and the simulation results. In response, we have performed a goodness-of-fit analysis for all experimental conditions presented in the new Fig. S11. Specifically, we calculated two metrics: the Kolmogorov-Smirnov (K-S) statistic (D) and the Root Mean Square Error (RMSE). The K-S statistic quantifies the maximum vertical deviation between the experimental and simulated cumulative distribution functions, while the RMSE assesses the overall deviation across the time course.

Fig. S11 has been updated to explicitly visualize these metrics; a vertical black line segment

now indicates the position and magnitude of the K-S statistic (D) for each condition, and the corresponding RMSE values are listed in the insets. The analysis yielded an average K-S statistic of $D_{\text{avg}} = 0.17$ across the 11 tested conditions. While visual deviations are indeed apparent in certain regimes, we consider this level of agreement satisfactory given the complexity of the global fitting challenge. Unlike local fitting, our model employs a single, global set of rate constants to describe the system's behavior across RNA and enzyme concentrations. The observed deviation of $\sim 17\%$ reflects the trade-off between minimizing local fitting errors and establishing a physically consistent mechanism that holds true across all experimental conditions. We have also detailed the methodology used for these calculations in Supplementary Text 3 and highlighted the relevant sections in the revised manuscript.

Minor issues:

1. In Supplementary Data 1-3, the timelapse videos recorded using fluorescence microscopy should display a scale bar. In Supplementary Data 3, the colloidal particle appears larger than in previous videos, suggesting that a different magnification was used. It would be helpful for the authors to clarify or standardize the imaging parameters.

We thank the reviewer for this careful review. We have added scale bars to the timelapse videos in Supplementary Data 1–3 as requested. regarding the appearance of the colloidal particle in Supplementary Data 3, we would like to clarify that the magnification remained consistent across all videos. The apparent increase in particle size is not due to a change in optical parameters, but rather a result of contrast adjustment performed using ImageJ (as detailed in the "Data recording and particle tracking" section of the SI). This adjustment was necessary to sufficiently enhance the particle's brightness, ensuring that its position could be accurately detected and tracked by the trackpy package. Consequently, while the particle appears larger due to the enhanced signal intensity and contrast, this is a visual artifact of the image processing required for data analysis, not a physical change or a difference in magnification.

2. The caption for Table S3 should be updated as the origami design created in caDNA_{no} has been moved from Fig. S11 to S2.

Thank you for noting this. We have updated the caption for Table S3 to correctly reference Figure S2.

3. The caption for Fig. 4L is incorrect. The graph plots τ_O and τ_C against enzyme concentration, not RNA concentration.

Thank you for catching this. We have corrected the caption for Figure 4L. It now correctly states that τ_O and τ_C are plotted against enzyme concentration, rather than RNA concentration.

4. In the revised version of the Supplementary Information, Supplementary Text 2 (“Simulation of the τ_C of the RNA-fueled engine”) and 3 (“Simulation of the τ_O of the RNA-fueled engine”) still refer to the previous figure numbering. These should be updated to match the current figure order.

a) The three-state model of RNA hydrolysis by RNase H enzyme is now illustrated in Fig. S8 (not S6).

b) Cumulative distribution functions (CDFs) of the folded-state dwell times (τ_C) at various enzyme concentrations are in Fig. S9 (not S7).

c) The kinetic pathway model describing the dwell time in the unfolded state is in Fig. S10 (not S8).

Thank you for the careful scrutiny. We have updated the figure references in Supplementary Text 2 and Supplementary Text 3 to align with the current figure order in the revised Supplementary Information.

5. Similarly, the caption for Fig. S12 should be updated to reference the correct figure numbers.

We have updated the caption for Figure S12 to reference the correct figure numbers. The caption now reads: "The simulation results were generated using the kinetic model described in Fig. S8 and S10, with parameters extracted from the fits in Fig. S9 and S11."

Reviewer #2 (Remarks to the Author):

We are grateful for your contribution and for participating in the Nature Communications co-review initiative that recognizes Early Career Researchers.

Reviewer #3 (Remarks to the Author):

I think the authors adequately addressed my comments and those of other reviewers. I support publication, but request the following minor additional revisions for better clarity and correction of errors.

Major points

1. Page 3, Lines 132 and 134. “double -exponential” should be “double-exponential”.

We thank the reviewer for pointing this out. We have corrected the hyphenation to “double-exponential” on Page 3.

2. Page 3, Line 133. “exponential(Fig. S6” should be “exponential (Fig. S6”.

We have added the missing space before the parenthesis.

3. Page 4, Lines 135 and 152. “the dwell times” should be “the mean dwell times” for better clarity.

We agree with the reviewer’s suggestion. We have updated the text to “the mean dwell times” on Page 4 to improve clarity.

4. Page 4, Line 142. Please add the following sentence after “(Fig. S7B).” to explain “a scrambled RNA linker” for better clarity.

The scrambled linker preserves the length but was sequence-shuffled to eliminate complementarity to both sticky ssDNA ends (Table S1).”

We have inserted the suggested sentence on Page 4 to clarify the design of the control. The text now reads: “The scrambled linker preserves the length but was sequence-shuffled to eliminate complementarity to both sticky ssDNA ends (Table S1).”

5. Page 5, Line 181. “hybridization(Fig. 4L).” should be “hybridization (Fig. 4L).”

We have added the necessary space.

6. Page 5, Line 220. “about10” should be “about 10”.

We have corrected the spacing error to read “about 10”.

7. Page 6, Line 266, and Page 7, Line 272. “Information..” should be “Information.”

We have removed the redundant periods.

8. Supplementary Materials, Page 5, Line 154. “AICdoubleand” should be “AICdouble and”.

We have corrected the typo to separate “AICdouble” and “and”.

9. Supplementary Materials, Page 5, Line 167. “{ti}with” should be “{ti} with”.

We have added the missing space.

10. Supplementary Materials, Page 8, Fig. S2. Please enlarge the sequence names in this figure for better clarity.

We appreciate this suggestion. We have modified Figure S2 in the Supplementary Materials to

enlarge the sequence names for better readability.

11. Supplementary Materials, Page 10, Line 254. “Table S3..” should be “Table S3.”

We have deleted the extra period after “Table S3”.

12. Supplementary Materials, Page 12, Line 267. “CDF(t)from” should be “CDF(t) from”.

We have corrected the spacing to read as “CDF(t) from”.

13. Supplementary Materials, Page 22, Line 351. “Fig. S11” should be “Fig. S2”.

We thank the reviewer for catching this error. We have updated the reference to “Fig. S2”.

REPLY TO REVIEWERS

Reviewer #1 (Remarks to the Author):

The authors have now addressed the comments and critiques.

We thank the reviewer for their positive feedback and for their time and effort in reviewing our manuscript. We are pleased that our revisions have satisfactorily addressed the comments.

Reviewer #2 (Remarks to the Author):

We are grateful for your contribution and for participating in the Nature Communications co-review initiative that recognizes Early Career Researchers.